# Influence of fuel ethanol content on primary emissions and secondary aerosol formation potential for a modern flex-fuel gasoline vehicle

Hilkka Timonen[1], Panu Karjalainen[2], Erkka Saukko[2,3], Sanna Saarikoski[1], Päivi Aakko-Saksa[4], Pauli Simonen[2], Timo Murtonen[4], Miikka Dal Maso[2], Heino Kuuluvainen[2], Matthew Bloss[1], Erik Ahlberg[5,6], Birgitta Svenningsson[6], Joakim Pagels[7], William H. Brune[8], Jorma Keskinen[2], Douglas R. Worsnop[9], Risto Hillamo[1], Topi Rönkkö[2]

[1] Finnish Meteorological Institute, Atmospheric Composition Research, P.O. Box 503, FI-00101 Helsinki, Finland

[2] Tampere University of Technology, Faculty of Natural Sciences, Aerosol Physics, P.O. Box 692, FI-33101 Tampere, Finland

[3] Currently at: Pegasor Oy, Hatanpään valtatie 34C, FI-33100 Tampere, Finland

[4] VTT Technical Research Centre of Finland, P.O. Box 1000, FI-02044 VTT, Finland

[5] Centre for Environmental and Climate research, Lund University, Box 118, SE-22100 Lund, Sweden

[6] Division of Nuclear Physics, Lund University, Box 118, SE-221 00 Lund, Sweden

[7] Division of Ergonomics & Aerosol Technology, Lund University, Box 118, SE-22100, Lund, Sweden

[8] Department of Meteorology, Pennsylvania State University, University Park, PA, US

[9] Aerodyne Research Inc., Billerica, MA, USA

*Corresponding author: Hilkka Timonen (hilkka.timonen@fmi.fi)

**Abstract**

The effect of fuel ethanol content (10%, 85%, 100%) on primary emissions and on subsequent secondary aerosol formation was investigated for a EURO5 flex-fuel gasoline vehicle. Emissions were characterized during a New European Driving Cycle (NEDC) using a comprehensive setup of high time resolution instruments. Detailed chemical composition of exhaust particulate matter (PM) was studied using a soot particle aerosol mass spectrometer (SP-AMS) and secondary aerosol formation using a potential aerosol mass (PAM) chamber. For the primary gaseous compounds, an increase in total hydrocarbon emissions and a decrease of aromatic BTEX (benzene, toluene, ethylbenzene and xylenes) compounds was observed when the amount of ethanol in fuel increased. In regard to particles, the largest primary particulate matter concentrations and potential to form secondary particles were measured for the E10 fuel (10% ethanol). As the ethanol content of the fuel increased, a significant decrease in average primary particulate matter concentrations over the NEDC cycle was found, PM emissions being 0.45, 0.25 and 0.15 mg m$^{-3}$ for E10, E85 and E100, respectively. Similarly, a clear decrease in secondary aerosol formation potential was observed with larger contribution of ethanol in fuel. Secondary to primary PM ratios were 13.4, and 1.5 for E10 and E85, respectively. For E100 a slight decrease in PM mass was observed after the PAM chamber, indicating that the PM produced by secondary aerosol formation was less than the PM lost via e.g. wall losses or degradation of primary organic aerosol (POA) in the chamber. For all fuel blends, the formed secondary aerosol consisted mostly of organic compounds. For E10 the contribution of organic compounds containing oxygen increased from 35%, measured for primary organics, to 62% after the PAM chamber. For E85 the contribution of organic compounds containing oxygen increased from 42 % (primary) to 57% (after the PAM chamber), whereas for E100 the amount of oxidized organics remained the same (approximately 62%) with the PAM chamber when compared to the primary emissions.

## 1. Introduction

Vehicular engine emissions are known to degrade air quality in urban areas. Besides gaseous compounds (e.g. CO, $NO_x$, hydrocarbons, volatile organic compounds), vehicle exhaust contains significant amounts of primary particulate matter (PM) (e.g. Maricq, 2007; Keuken et al., 2013; Gordon et al., 2014a). Primary particulate matter refers to particles directly emitted e.g. from engine, fuel combustion process or brakes, and not yet experienced any significant chemical transformation in the atmosphere. Depending on engine and fuel type, primary exhaust PM emissions from vehicles consist mainly of soot and different fuel and lubricating oil components (Maricq, 2007; Canagaratna et al., 2010; Karjalainen et al., 2014). In addition to primary PM, burning processes in engine cylinders produce so called delayed primary aerosol. Delayed primary include species like sulfuric acid which occur in tailpipe conditions in gaseous phase but will condense or nucleate immediately when the exhaust is cooled and diluted, without any significant chemical transformation in the atmosphere (Arnold et al., 2012; Rönkkö et al., 2013; Pirjola et al., 2015). In particle number size distribution, the exhaust PM formed by these different processes are frequently seen as separate modes with different concentrations and particle size ranges (Kittelson, 1998; Rönkkö et al., 2013). In addition to primary emissions, large amounts of secondary particulate matter forms after the exhaust gases are released into the atmosphere (Chirico et al., 2010). Secondary particulate matter forms in the atmosphere via gas-to-particle conversion as oxidation processes typically lowers the volatility (vapour pressure) of gaseous compounds. The difference between delayed primary and secondary emission is that secondary emissions form through different transformation processes in the atmosphere, whereas delayed primary emissions form in the cooling process without any significant chemical transformation due to external conditions such as ultraviolet light (UV) or atmospheric oxidants. While a large number of studies have focused on vehicular primary particulate emissions (Giechaskiel et al., 2005; Maricq, 2007; Lähde et al., 2010; Karjalainen et al., 2014) a relatively limited number of studies have focused on secondary emissions.

Both batch chambers (such as smog chambers) and flow through chambers combined with modern online composition analysis (e.g. AMS) have been used to study vehicular secondary aerosol emissions in both laboratory and ambient conditions. Smog chambers have been used to study e.g. composition of primary and secondary PM in exhaust emissions of gasoline and diesel vehicles, influence of after treatment to secondary aerosol formation for diesel vehicle as well as fraction of the emissions that form secondary organic aerosol (SOA) and the relative importance of primary PM emissions versus SOA formation (e.g. Nordin et al., 2013, Platt et al., 2013, Chirico et al., 2014, Gordon et al., 2014, Presto et al., 2014). A batch chamber is good for detailed oxidation process studies (e.g. Chirico et al., 2014; Suarez-Bertoa et al., 2015), but cannot be used to differentiate rapidly changing driving conditions during the test driving cycle. Flow through chambers, such as the Potential Aerosol Mass (PAM) chamber, are designed to simulate secondary aerosol mass formation potential on a close to real-time basis (Kang et al., 2011; Lambe et al., 2011). Several studies have been recently published where the PAM chamber was used to study vehicular emissions from gasoline, diesel and flex-fuel vehicles (e.g. Kroll et al., 2012, Suarez-Bertoa et al., 2015, Karjalainen et al., 2016; Jathar et al., 2017). These studies have shown that secondary particulate emissions from the combustion engines mainly consists of organic compounds and ammonium nitrate (Karjalainen et al., 2016; Suarez-Bertoa et al., 2015) and that the secondary PM emissions can be significantly larger than primary emissions if conditions favours secondary aerosol formation (Giechaskiel et al., 2005; Chirico et al., 2010; Karjalainen et al., 2015). In gasoline vehicles exhaust emissions of secondary aerosol precursors have been shown to depend e.g. on driving conditions, fuels and the operation of catalytic converters (Durbin et al., 2007; Maricq et al., 2012; Gordon et al., 2014b; Karjalainen et al., 2016). Also, previous studies indicate that gasoline vehicles have an impact on secondary aerosol concentrations in urban areas (Nordin et al., 2013; Tkacik et al., 2014; Karjalainen et al., 2016; Suarez-Bertoa et al., 2015). However, the secondary aerosol formation potential and composition as a function of driving situation for different ethanol content fuels (E10-E100) remain poorly characterized in the literature.

The hydrocarbons of gasoline typically includes 4-12 carbon atoms with boiling range between 30 and 210 °C (Owen and Coley 1995). These may be present in the exhaust gases as unburned hydrocarbons. In addition, exhaust gases contain compounds formed in combustion and those originating from engine oil. Lipari et al. (1990) analysed 103 individual hydrocarbons up to C12 in a study with flex-fuel vehicles (FFV) using gasoline and methanol containing fuels M85 and M100 (85, 100% methanol). For gasoline, toluene, ethylene, propylene, isobutylene, isopentane, pentane, benzene and iso-octane represented 55% of total hydrocarbons. These gaseous compounds are emitted into the atmosphere directly or they are evaporated from primary exhaust particles when the exhaust is diluted (Robinson et al., 2007). Oxidation products of organic compounds may contain one or more functional groups such as alcohol (-OH), aldehyde (-CHO), carboxylic acid (–COOH), nitro ($-NO_2$) and nitrate ($-NO_3$) or organic sulfate ($-OSO_3$) groups. Ambient photochemical reactions take place in the presence of $NO_2$, volatile organic compounds (VOCs), heat and sunlight. Hundreds of different VOC species can participate in thousands of photochemical reactions (Drechsler 2004). Different possible photo-oxidation pathways are also dependent on conditions. For aromatic BTEX (BTEX = benzene, toluene, ethylbenzene, and xylenes; VOCs typically found in petroleum derivates) compounds have been suggested to depend on, for example, prevailing $NO_x$ concentrations during the aging process (Andino et al., 1996; Hurley et al., 2001; Sato et al., 2007; Sato et al., 2012).

The European Union has set an obligation that the share of renewable energy should be at least 10% in the transportation sector by 2020 (Directive 2009/28/EC). Ethanol is the dominant bio-component in transport fuels worldwide. However, in Europe its share in gasoline is limited to 10 vol-%, which is equivalent to approximately 6% energy content [Directive 2009/30/EC]. Higher ethanol concentrations up to 85 vol-% (E85) can be used in special flex-fuel vehicles. Previous studies have shown that primary PM, CO, HC, $NO_x$ and aromatic hydrocarbon emissions are typically lower for the E85 fuel than for gasoline, whereas ethanol, acetaldehyde, formaldehyde and methane emissions increase with increasing ethanol content of gasoline (Yanowitz et al. 2013; Nylund and Aakko, 2003; Karlsson et al., 2008; Westerholm et al., 2008; Clairotte et al., 2013). In order to reduce the detrimental effects of pollution caused by vehicles, the emission standards for PM emissions of vehicles are getting tighter globally. However, it must be noted that in the emission standard laboratory tests the PM mass is measured directly after the tailpipe from a filter sample at elevated temperature and thus represents mainly primary non-volatile PM emissions. As the previous studies have demonstrated (e.g. Chirico et al., 2010; Nordin et al., 2013; Platt et al., 2013; Suarez-Bertoa et al., 2015), the secondary PM emissions, formed from gaseous precursors, can be significantly larger than primary PM emissions, meaning that the emission limits do not necessarily regulate secondary PM emissions.

In order to properly quantify vehicular engine emissions, the whole transformation chain from freshly emitted primary PM and gaseous compounds to aged secondary PM measured in urban air quality stations has to be better understood. The main objective of this study was to investigate primary particulate emissions and simulate the secondary aerosol formation potential of vehicular emissions with an oxidation flow chamber when the ethanol content in fuel increases. Measurements were carried out with a modern FFV using fuels with three different ethanol contents (10%, 85% and 100%). A comprehensive set of instruments were used for measuring gaseous emissions together with chemical composition and size distributions of primary and secondary particles. All measurements were done with high time resolution instruments and with the PAM flow-through oxidation chamber. The measurement setup used, enabled the characterization of concentration and composition changes during different parts of the driving cycle.

## 2. Experimental

### 2.1 Measurement setup and sampling

The measurement setup of this study is described in detail by Karjalainen et al., 2016. The article made by Karjalainen et al., 2016 is focused on primary and secondary emissions of a flex-fuel vehicle using E10 fuel, whereas this article is focused on influences of fuel alcohol content to particulate and gaseous emissions and their composition. In this study, emissions from a flex-fuel passenger car (model year 2011, 1.4 litre turbo charged direct injection spark ignition (DISI) engine, EURO5) were measured on a chassis dynamometer at 23°C using three different fuels (E10, E85, E100; gasoline with 10, 85 and 100% alcohol). Schematic figure of measurement setup shown in Fig. S1. The FFV vehicle was conditioned according to the manufacturer's instructions, and the adaptation of the car to new fuel was monitored. Preparation needs and stability issues related to the FFV cars were based in the earlier project (Aakko-Saksa et al., 2014). The driving cycle was New European Driving Cycle (NEDC; cycle profile shown in Fig. S2). NEDC totals 11.0 km, divided into three test phases to study emissions at cold start and with a warmed-up engine. First part of the NEDC cycle, the urban driving cycle (UDC) is repeated twice. The first phase, CSUDC, represents urban driving with a cold start (0-391 s, Cold Start UDC), the second phase, HUDC, represents typical urban driving (392-787s Hot start UDC) and the last phase, EUDC represents highway driving (788-1180s, Extra-urban driving cycle). NEDC cycles were run on separate days in order to enable cold start conditions for each fuel. Test fuels comprised of a regular commercial E10 (max 10% ethanol), E85 (85% ethanol), and E100 (100% ethanol). To avoid engine problems related to lean ethanol, deionized water was added into E100 to adjust water content to 4.4 % (m/m). A more detailed description of car preparation and driving cycle is given in the supplemental material. Particle sampling was conveyed by partial exhaust sampling system (Ntziachristos, 2004) at the exhaust transfer line. The sampling system consisted of a porous tube diluter (PTD, dilution ratio DR = 12), residence time chamber (2.5 s) and secondary dilution conducted by a Dekati Diluter (DR = 8). Regarding particle formation by nucleation, the sampling system mimics exhaust dilution and nanoparticle formation processes in the atmosphere (Rönkkö et al., 2006; Keskinen and Rönkkkö, 2010). Two NEDC tests were conducted for each fuel. While some parameters were monitored similarly during both of these (gaseous emissions, particle size distribution of primary exhaust particles; shown in Table S1), the extensive study for the differences between primary and secondary particle emissions could only be conducted once per fuel.

The PAM chamber was used to evaluate secondary aerosol formation potential during the NEDC driving cycle. The PAM chamber is a small, flow-through chamber that is irradiated with ultraviolet light (wavelengths 185 and 254 nm) to form high concentrations of oxidants ($O_3$, OH, $HO_2$) that can initiate the production of secondary aerosol particles (Kang et al., 2007; Kang et al., 2011, Lambe et al., 2011). High oxidant concentrations (up to 1000-fold to atmosphere, with the same oxidant ratios as in the atmosphere) and high UV lights assure the fast oxidation of compounds (Kang et al., 2007). The aging as the sample flows through the chamber is shown to represent up to several weeks of aging in the atmosphere (Kang et al., 2011; Ortega et al., 2013). The PAM chamber has been thoroughly characterized in previous studies. These studies include loss characterization, comparison to other chambers studies as well as comparison on how SOA formed in chamber compares to SOA observed in ambient atmosphere and SOA produced in large environmental chambers as well (e.g. Kang et al., 2007; Kang et al., 2011; Lambe et al., 2011; Tkacik et al., 2014; Lambe et al., 2015; Peng et al., 2016). The PAM chamber used in this study is described in detail by Karjalainen et al., 2016. Shortly, the PAM chamber was installed between the residence time chamber and secondary dilution unit of sampling system (Fig. S1). The particle instrumentation was located downstream of the secondary diluter. The sample flow through the PAM chamber was set to 9.75 l min$^{-1}$ resulting in an average residence time of 84 s. The voltage of the two UV lamps was at maximum value, 190 V. The sample conditions during the test were fairly stable, typically relative humidity was 60%, temperature 22 °C and ozone concentration 6 ppm. All cycles were first run without the PAM chamber to measure primary emissions and next with the PAM chamber in order to study the formation of

secondary particles. The secondary aerosol in PAM chamber is formed when low volatility vapors condense on aerosols or form new particles. In the PAM chamber, these vapors may also condense onto walls, exit the chamber, or react with OH, which leads to fragmentation and an increase in the saturation vapor pressure. Thus the potential aerosol mass is underestimated if these chamber related losses of low volatile vapors are not taken into account. We used the LVOC (low volatility organic compound) fate model presented by Palm et al. (2016) to estimate the losses of condensing organic vapors in the PAM chamber (model available at https://sites.google.com/site/pamwiki/hardware/estimation-equations). PM losses in the chamber were studied in the laboratory using a similar PAM chamber as in the measurements. Supplemental material includes a detailed description of loss calculations and measured PM losses as a function of particle size. Shortly, losses of primary PM in a PAM chamber (Fig. S3) are in general small especially in the particle sizes that contain most of the aerosol mass: 25% at 50 nm, 15% at 100nm and below 10% above 150 nm. Also, because of the high condensational sink, over 95 % of the LVOCs condensed on aerosol in all cases according to LVOC fate model. Thus, the chamber related losses of LVOCs and PM are small.

The PAM chamber was calibrated following the procedure described by Lambe et al. (2011). According to this off-line calibration, the upper limit average OH exposure during the experiments was $1.0 \times 10^{12}$ molec. cm$^{-3}$ s, corresponding to atmospheric aging of 8 days (assuming an average OH concentration of $1.5 \times 10^6$ molec. cm$^{-3}$ in the atmosphere (Mao et al., 2009)), but the real OH exposure is lower due to high concentrations of OH reactive gases in the exhaust. This effect is significant especially at the beginning of the cycle (CSUDC) when the concentrations are high. The average external OH reactivities (OHR) due to VOCs and CO in CSUDC were 1246 s$^{-1}$, 1141 s$^{-1}$ and 3441 s$^{-1}$ for E10, E85 and E100, respectively. The higher OHR of E100 is due to high ethanol and aldehyde concentrations. After the CSUDC, the average OHR is below 90 s$^{-1}$ for all fuels. More detailed OHR calculations are shown in Tables S2-S4.

We estimate the OH exposure in the PAM by using a simple photochemical box model made by William Brune, in which the differential equations describing the chemical reactions are solved using Euler's method. More details and the source code of the model are found in PAM users manual (https://sites.google.com/site/pamusersmanual/7-pam-photochemistry-model). The free parameters in the model are photon fluxes at 254 nm and 185 nm wavelength. Based on the off-line calibration, the best-fit values for the photon fluxes are $7.3 \times 10^{14}$ photons cm$^{-2}$ s$^{-1}$ and $1.3 \times 10^{13}$ photons cm$^{-2}$ s$^{-1}$ for 254 nm wavelength and 185 nm wavelength, respectively. The inputs for the model are OHR due to VOCs, CO concentration, NO concentration and $NO_2$ concentration. In the model, $SO_2$ is used as a proxy for VOCs, i.e,. in the model, the OHR of $SO_2$ equals the input OHR due to VOCs. This method is reasoned to be a realistic approximation by Peng et al. (2015) in terms of estimating the OH exposure.

The input values for the model are obtained from 1-second time resolution measurements of CO, $NO_x$ and total hydrocarbons (THC), corrected with the residence time distribution caused by the PAM chamber. The residence time distribution is obtained from the $CO_2$ pulse experiment presented by Lambe et al. (2011). The concentrations of individual VOCs are estimated using the high time-resolution THC concentration and the distribution of VOCs in different phases of the driving cycle (see Tables S2-S4), and the OHR due to VOCs is obtained from these concentrations and respective reaction constants. The OH exposure in PAM was modelled at a 20 second time interval for each driving cycle, and the average OH exposures for the cycles are presented in Table 1.

## 2.2. Particle measurements

The Soot Particle Aerosol Mass Spectrometer (SP-AMS, Aerodyne Research Inc., US) was used to measure the chemical composition of emitted PM. The SP-AMS is a high resolution time-of-flight aerosol mass spectrometer (HR-ToF-AMS) with added laser (intracavity Nd:YAG, 1064 nm) vaporizer. The dual vaporizer system enables the real-time measurements of PM mass and size-resolved chemical composition of submicron non-refractory particulate matter, refractory black carbon and some metals and elements (e.g. Na, Al, Ca, V, Cr, Mn, Fe, Ni, Cu, Zn, Rb, Sr and Ba; Carbone et al., 2015). The HR-ToF-AMS is described in detail by Jayne et al. (2000) and DeCarlo et al. (2006), and the design of the SP-AMS by Onasch et al. (2012). Briefly, in the SP-AMS an aerodynamic lens is used to form a narrow beam of particles that is transmitted into the detection chamber. Particles are vaporized either by tungsten vaporizer (600 °C) to analyze non-refractory inorganic species and organics and/or with the laser in order to analyze refractory black carbon (rBC) and metals in addition to inorganics and organics attached to these particles. The vaporized compounds are ionized using electron impact ionization (70 eV) and formed ions are guided to the time-of-flight chamber and to the multi-channel plate (MCP) detector. A five second averaging time and dual-vaporization system, with both laser and tungsten oven operating, was used in the measurements. Only V-mode data is used in this study. For SP-AMS one second and 1 minute $3\sigma$ detection limits for submicrometer aerosol are $<0.31$ µg m$^{-3}$ and $< 0.03$ µg m$^{-3}$ for all species in the V-mode, respectively (DeCarlo et al., 2006, Onasch et al., 2012). $CO_2$ concentrations during the measurements were significantly higher (up to 1450 ppm) than atmospheric values, thus $CO_2$ time series was used to correct the artefact caused by gaseous $CO_2$.

The collection efficiency (CE) value, representing the fraction of sampled particle mass that is detected by the MCP detector, is required for the calculation of aerosol mass concentration measured by the AMS. The previous studies have shown that the collection efficiency of an aerosol mass spectrometer is affected by particle losses (i) during transit through the inlet and lens, (ii) by particle beam divergence for both tungsten and laser vaporizers and by (iii) bounce effects from tungsten vaporizer (Matthew et al., 2008; Huffman et al., 2009; Onasch et al., 2012). Willis et al. (2014) demonstrated that also particle morphology affects the SP-AMS particle beam width that in turn affects the collection efficiency through the overlap of the particle beam and the laser beam. Similar to Karjalainen et al. (2015) a CE = 1 was used in this study for all SP-AMS data. We acknowledge that it is likely that the collection efficiency might be underestimated for thinly coated, primary emissions whereas used CE = 1 is likely closer to the correct value for heavily coated spherical secondary aerosol. Also, we note that gasoline soot, consisting of agglomerates with average diameter below 90nm, will likely have low transmission efficiency in the aerodynamic lens and thus might have lower collection efficiency than regal black, that is typically used for calibration.

The particle number size distributions were measured using a time resolution of 1 Hz with a High-Resolution Low-Pressure Cascade Impactor (HR-LPI; Arffman et al., 2014) and an Engine Exhaust Particle Sizer (EEPS, TSI Inc.; Mirme, 1994; Johnson, 2004). The particle number concentration was also measured with an ultrafine condensation particle counter (UCPC, TSI Inc. model 3025). The UCPC was located downstream of an additional diluter (operation principle based on the partial filtration of the sample, DR = 42) to ensure that the concentrations to be measured are within its measurement range. All the data shown below has been corrected by a total dilution ratio for each instrument, thus the presented values represent the tailpipe concentrations.

The particle number size distributions measured by HR-LPI can be used to estimate how the particle losses in the PAM affect the measured total particle mass. If the measured HR-LPI number size distributions are corrected with the particle loss curve (Fig. S3), the total mass calculated from the number size distribution increases by 9-16 % depending on the phase of the cycle and the fuel (see Table S5 for details). The masses measured by the SP-AMS cannot be corrected in a similar way, since the SP-AMS did not measure the particle size distributions and different chemical species might be located in different-sized

particles. Thus, the SP-AMS results presented in the following sections are not corrected for the particle losses in the PAM, but we expect that the loss of e.g. organic mass due to PAM wall-losses is of similar order as the loss of total HRLPI mass.

## 2.3. Gaseous phase composition measurements

Total hydrocarbon (THC) concentrations were measured with a flame ionization detector (FID) developed for the standardized exhaust emission test procedures of cars. The FID detects all carbon-containing compounds, for example carbonyl compounds, in addition to hydrocarbons (HC; Sandström-Dahl et al., 2010, Aakko-Saksa et al. 2014). In addition, samples were collected using Tedlar bags for subsequent analysis by a gas chromatograph (HP 5890 Series II, AL2O3, KCl/PLOT column, an external standard method). The analysed hydrocarbons (from $C_1$ to $C_8$) included methane, ethane, ethene, propane, propene, acetylene, isobutene, 1,3-butadiene, benzene, toluene, ethyl benzene and m-, p- and o-xylenes.

Besides HCs, selected aldehydes were analysed by collecting diluted exhaust gas samples from the Constant Volume Sampler (CVS) using 2,4-dinitrophenylhydrazine (DNPH) cartridges. The DNPH derivatives were extracted with an acetonitrile/water mixture and analysed using HPLC (High Performance Liquid Chromatography) technology (Agilent 1260, UV detector, Nova-Pak C18 column). Aldehydes analysed include formaldehyde, acetaldehyde, acrolein, propionaldehyde, crotonaldehyde, methacrolein, butyraldehyde, benzaldehyde, valeraldehyde, m-tolualdehyde and hexanal. Ethanol and a number of other compounds were measured on-line using Fourier transformation infrared (FTIR) analyzer (Gasmet Cr-2000). A summary of measured gaseous compounds, used instrument and their detection limits is shown in table 2.

In these measurements, the sum of hydrocarbons (HCs) analysed by GC, FTIR and HPLC (sum of HC from GC + HC portions of ethanol and acetaldehyde) resulted in HC sum of 15 mg/km for E10, 30 mg/km for E85 and 216 mg/km for E100. The respective THC (FID) results were 22 mg/km, 30 mg/km and 193 mg/km. This indicated that on average 73% of THC (FID) were analysed for E10, and 100 % for E85 and E100 by the GC, FTIR and HPLC analyses.

## 3. Results

### 3.1. Gas phase emissions

The composition of gas phase emissions was observed to change when the ethanol content of the fuel changed (Fig. 1). As the ethanol content increased a clearly detectable decrease was observed in both average $NO_x$ and ammonia concentrations during the measurement cycle. Decrease in $NO_x$ with increasing ethanol content is likely caused by decreased flame temperature (Turner et al., 2011). For instance Turner et al., 2011 reported a decrease of about 20 to 40 °C in exhaust temperature of a DISI engine when the ethanol content of fuel changed from 0 to 100%. Simultaneously, they reported that the $NO_x$ emission decreased from 8 g/kWh to 0.5 g/kWh, both values slightly depending also on ignition timing and strategies. Decreasing trend in the $NO_x$ emissions have also been observed by Maricq et al. (2012) who report the decreases of 20% in $NO_x$ emissions when the ethanol content increases to values more than 17%.

Ammonia is formed in the reactions of the three-way catalyst (TWC; Mejia-Centeno et al., 2007). In theory, ammonia formation is enhanced in slightly rich air-to-fuel ratio at high temperatures (aggressive accelerations) when sufficient HC and $NO_x$ concentrations are present (Heeb et al., 2006; Mejia-Centeno et al., 2007; Li et al., 2010). Engine-out emissions were not measured here, but it is assumed the HC and $NO_x$ concentrations were not a limiting condition for ammonia formation. Decrease in the ammonia emissions for the E85 fuel indicates an enleanment of the conditions in the TWC catalyst when compared with those for the E10 fuel. Clairotte et al. (2013) also reported lower ammonia emissions for the E85 fuel than for

the E5 fuel. The decreases in both ammonia and $NO_x$ leads to a decreased contribution to secondary aerosol formation of ammonium nitrate in the atmosphere when increasing the ethanol content.

The amount of hydrocarbon emissions typically depends on combustion conditions and exhaust after-treatment by catalytic devices. In this study, the test vehicle was equipped with a three-way catalytic converter, with an effectivity which depends on exhaust temperature and also on hydrocarbons properties. In this study the composition of hydrocarbon emissions were observed to be strongly dependent on the ethanol content; as the ethanol content increased in fuel, short chain non-aromatic hydrocarbons and aldehydes increased in the exhaust, while a decreasing trend was observed for all measured aromatic hydrocarbon compounds. Also as the ethanol content of fuel increased, the exhaust concentrations of formaldehyde, acetaldehyde, ethanol, methanol, ethene and acetylene increased whereas the exhaust concentrations of $NO_x$, ammonia, PM and BTEX decreased (Fig. 1).

### 3.2. Composition of primary particulate matter emissions

The chemical composition of primary particulate emissions was observed to vary for different fuels (Fig. 2). Concentration for each chemical component in units mg/km for both primary and secondary emissions is shown in the supplement Fig S4. For E10 approximately half of primary PM emissions was rBC and the other half was organics. The contribution of inorganic species (sulfate, nitrate, ammonium, chloride) to PM mass was small (1.2 %). From inorganic ions, sulfate had the highest contribution (47–67% of mass of inorganic ions) for all fuels. A clear decrease in rBC concentration and in its contribution to total emitted primary PM was observed as the ethanol content of fuel increased (E10, rBC: 53%, E85, rBC: 31%, E100, rBC: 25%), whereas contribution of organic matter increased from 46% (E10) to 65% for E85 and 75% for E100. Organics to rBC –ratios for the E10, E85 and E100 were 0.9, 2.1 and 3.1, respectively.

Figure 3a shows the elemental ratios (oxygen to carbon ratio: O:C and hydrogen to carbon ratio: H:C) for primary organic PM. All values are average values over the NEDC cycle. Average elemental composition and elemental ratios (O:C, H:C) are calculated using a method developed by Aiken et al. (2007) where elemental composition is calculated using all measured fragment ions observed in high-resolution mass spectra and H:C, O:C calibration factors derived from laboratory measurements of standard organic molecules. Canagaratna et al. (2015) further developed elemental analysis to contain a wider range of organic species that are more representative of ambient organic aerosol (OA) species. Improved ambient ratios (IA) calculated according to method published by Canagaratna et al. (2015) are also shown in Figure 3. Rather similar O:C values (0.35–0.4) were observed for the primary organic fraction for all fuels. Observed H:C for primary emissions of E10 and E85 (~1.5) was slightly lower than for E100 (1.7). Observed elemental ratios are comparable to elemental ratios typically observed for hydrocarbon-like organic aerosol (HOA) representing traffic emissions in ambient atmosphere and elemental ratios observed in chamber studies (Tkacik et al., 2012; Timonen et al., 2013; Carbone et al., 2014; Canagaratna et al., 2015). We note that these O:C measured with a SP-AMS can be slightly higher than ratios typically measured with an AMS for primary emissions from gasoline vehicles due to the fact that the SP-AMS also detects refractory organic species (rCOx) incorporated on the structures of black carbon particles (Corbin et al., 2014).

### 3.3. Composition of secondary aerosol simulated with a PAM chamber

Secondary aerosol formation in the PAM chamber increased the contribution of both organic and inorganic compounds (Fig. 4). For all fuels most of the particulate matter observed after the PAM chamber (E10; 89%, E85: 79%, E100: 61%) consisted of organic compounds. Organics to rBC –ratios for the secondary emissions of E10, E85 and E100 were 12, 8.3 and 3.1, respectively. The main inorganic ions observed after the PAM chamber were nitrate, sulphate and ammonium. When the ethanol content of fuel increased, the relative contribution of inorganic ions (E10: 4 %, E85: 12% , E100: 19 %) in exhaust PM increased after the PAM chamber.

### 3.3.1 Organic aerosol

Figure 3b shows the average elemental ratios over the NEDC cycle calculated for organics from the SP-AMS data. For E10 increase in O:C (from 0.4 to 0.6) and a decrease in H:C (from 1.55 to 1.45) was observed when the PAM chamber was used. In contrast, no change in O:C or H:C was observed for E85 when using the PAM chamber. For E100, a slight decrease in O:C and H:C values was observed when using the PAM chamber. Similar elemental ratios (O:C 0.4–0.7) (Nordin et al., 2013; Suarez-Bertoa et al., 2015) have been observed for secondary PM emissions in previous batch chamber studies. As shown for the gaseous exhaust compounds (chapter 3.1), the composition and concentrations of gaseous precursors change when the ethanol content of fuel increased, also causing a clear change in observed secondary aerosol composition and oxidation state.

Figure 5 shows the contribution of different organic fragments $C_xH_y$ (hydrocarbons with $C_xH_y^+$), $C_xH_yO$ (fragments with one oxygen atom $C_xH_yO^+$ e.g. $CO^+$ and $CHO^+$) and $C_xH_yO_z$ (hydrocarbon compounds containing several oxygen atoms $C_xH_yO_{z, z>1}^+$ e.g. $CO_2^+$) for all the fuels with and without the PAM. The contribution of $C_xH_yO^+$ to organics increased after the oxidation of exhaust sample in the PAM chamber for all the fuels, whereas the contributions of $C_xH_yO_{z, z>1}^+$ and $C_xH_y^+$ decreased. For E10 the contribution of the sum of oxidated compounds ($C_xH_yO^+$ and $C_xH_yO_z$) on exhaust PM was increased from 35% to 62%. For E85 the contribution of oxidized organic compounds increased from 42 to 57%, whereas for E100 the contribution of oxidized organic compounds (approximately 62%) remained the same with the PAM chamber when compared to primary emissions. For all fuels, the total contribution of oxidized compounds ($C_xH_yO$, $C_xH_yO_z$) increased in PAM chamber when compared to contribution of hydrocarbons. For E10, E85 the absolute concentration organic fraction and also total mass concentration of each organic hydrocarbon group increases in the PAM chamber, although the contribution of CxHyO slightly decreases as shown in the figure 5. Also, mass spectra (Fig. S5-S16) shows that in hydrocarbon composition clear differences can be observed. For E100, both contributions and concentrations of different organic families are similar with and without PAM chamber, however again a change in composition of these hydrocarbon groups is observed.

### 3.3.2 Refractive Black carbon

Refractive Black carbon (rBC) is formed during incomplete combustion and is always considered primary emission, therefore the rBC concentrations with and without the PAM chamber should be the same. Also, measurements of regulated emissions (Table S1) show that the cycles were repeatable and rBC concentrations for both cycles should be on same level. However, some differences in rBC concentrations were observed when primary rBC concentrations were compared to the rBC results measured after the PAM chamber. For E85 and E100 a slight decrease in rBC after the PAM chamber was observed (20-30%). This decrease is likely explained by losses of primary PM in the chamber. Karjalainen et al., 2016 showed that the particle losses in PAM chamber were on similar level as rBC losses seen for here (approximately 8-30% for particle sizes 50 nm-400 nm). However, in contrast, for E10 a clear increase in rBC (from 240 µg m$^{-3}$ to 480 µg m$^{-3}$) was observed when the emissions during driving cycle were measured with the PAM chamber. In the SP-AMS rBC is calculated as a sum of carbon fragments $C_x^+$. To explore the observed increase in rBC for E10 after the PAM chamber, the $C_x$ fragments before and after the PAM chamber were studied. The increase after the PAM chamber for E10 was seen in all $C_x$ fragments (Fig. S17; w PAM / w/o PAM –ratios from 1.6 to 2.8 for $C_2$-$C_5$ fragments). The main fragments $C_3^+$ and $C_2^+$ of rBC (contributions 59% and 27%, respectively) did not have any major interference from isobaric ions (ions observed in same nominal mass). Larger fragments (e.g. $C_5^+$-$C_9^+$) had interfering isobaric organic compounds, but their contribution to total mass was less than 10% and the influence of interference was therefore considered to be insignificant.

There can be several reasons why SP-AMS detected more rBC at the measurements conducted after the PAM chamber. Firstly, it must be noted that this increase in rBC was only observed for E10, which had the highest secondary aerosol formation

potential and thus largest increase in particle size in the PAM chamber. Previous studies have shown that the soot particles emitted by DISI vehicles are small, typically on the size range of 10-100 nm (Karjalainen et al., 2014). Due to restrictions from the aerodynamic lens, particles smaller than 50 nm are not effectively detected by the SP-AMS. The difference in the rBC results for E10 between primary emissions and emissions after the PAM is likely partly explained by increased mean particle size due to SOA formation increasing the efficiency through which particles are detected by the SP-AMS (low volatility compounds formed in the PAM chamber condense on the surfaces of soot particles increasing their aerodynamic size and thus the detection efficiency of soot/rBC). Also, Willis et al. (2014) demonstrated that the thick coating increases the collection efficiency via changing the particle morphology and thus decreasing the beam divergence and increasing the particle and laser beam overlap. Based on increased mean particle size and secondary to primary PM -ratios it can be assumed (see also chapter 3.4, and size distributions with and without PAM chamber figures S18, S19) that for E10 the soot particles are heavily coated with SOA after the PAM chamber and thus they are more effectively detected by the SP-AMS. In contrast, it has been shown that dispersion of small and nonspherical particles in the aerodynamic lens inlet of the SP-AMS may cause particles to miss the laser vaporizer (Onasch et al., 2012). Also, based on the previous studies, it is known that black carbon particles emitted by engines are typically agglomerates with irregular shape and diameter 10-90 nm (Happonen et al., 2010; Lähde et al., 2010; Karjalainen et al., 2014; Liati et al., 2016), which also might have decreased the detection efficiency of primary black carbon (soot) particles in this study. We also note that the losses for PM in the PAM chamber are dependent on particle size, Karjalainen et al., 2016 (Figure S3 in the supplemental material) found that the smallest particles incurred the largest losses in PM. However, based on this study, it is not possible to estimate which of these above mentioned processes is the main reason for observed rBC increase for E10.

### 3.3.3 Inorganic ions

In comparison to organics, observed inorganic ion concentrations after the PAM chamber were from moderate to low (ion contribution to PM mass E10: 3.8%, E85: 12% and E100: 19%). The main ions observed after the PAM chamber were sulphate, nitrate and ammonium for all fuels. In this study the sulphur contents of E10, E85 and E100 fuels were lower than 10 ppm according to the specifications EN 228, EN 15293 and EN 15376, respectively. These facts suggest strongly that, especially for E100, most of the observed sulphate emissions originates from lubricant oil.

### 3.4. Primary to secondary particulate matter -ratios

Table 3 and Figure 6 shows the submicron PM concentration for both primary emissions and for potential secondary aerosol emissions measured after the PAM chamber averaged over the driving cycle. PM was calculated as a sum of all SP-AMS species in the size range of the SP-AMS (30–800 nm). PM concentration measured after the PAM chamber is a sum of both primary particulate emissions and formed secondary aerosol. Secondary aerosol concentrations were calculated by subtracting the concentrations of primary particles from the PM concentrations observed after the PAM chamber. It is likely that wall losses in the chamber will somewhat decrease primary aerosol concentrations and thus might increase the observed secondary to primary PM –ratio when particles go through the PAM chamber. However based on laboratory PM loss measurements and modelled vapour losses, the influence of these to the results is estimated to be small. Also one has to note that it is likely that in the PAM chamber new material originating from the gaseous phase will condense on existing particles, which can change the particle morphology, increase the particle size and furthermore change their detection efficiency in the SP-AMS (Onasch et al., 2012; Willis et al., 2014) thus slightly also affecting observed secondary to primary PM ratios.

Large differences in the concentrations between primary and secondary emissions were observed for different ethanol content fuels. The largest primary and secondary PM concentrations were observed for E10 (0.45 mg m$^{-3}$ for primary PM and 6.7 mg m$^{-3}$ after PAM chamber). A clear decrease in primary PM emissions (E85; 0.24 mg m$^{-3}$, E100: 0.15 mg m$^{-3}$) were seen as the

ethanol content in fuel increased. Similar to our results, Maricq et al. (2012) observed a decrease in primary PM concentrations when ethanol content of fuel increased, however they did not measure the secondary aerosol formation potential. The ethanol content of fuel also had a large influence on the secondary aerosol formation potential. For E10 PM measured after the PAM chamber was on average 14.7 times larger than the primary PM. For E85 the secondary PM emissions after the PAM chamber were on average approximately two times larger (0.62 mg m$^{-3}$) than primary PM emissions and for E100 a slight decrease in PM mass (E100: 0.12 mg m$^{-3}$) was seen after PAM chamber, indicating that either the secondary aerosol formation was insignificant (Fig. 2, Table 3) or extensive fragmentation decreased observed PM mass. Previous studies (e.g. Tkacik et al., 2014) have shown that high OH exposures cause a reduction in the observed mass because of fragmentation, which forms light organic compounds that are more volatile and will evaporate from the particulate phase. In the case of E100, the fragmentation probably does not occur at the beginning of the cycle because of the low OH exposure. Corresponding secondary to primary PM ratios were 13.4 and 1.5 for E10 and E85, respectively.

The influence of fuel composition on secondary aerosol formation has been studied in only a few articles. Suarez-Bertoa et al. (2015) has studied secondary aerosol formation potential of exhaust for vehicles using high ethanol content fuels (E75, E85). They used a batch chamber in their study and found that the secondary aerosol, mostly secondary organic aerosol (SOA) was on average three times higher than primary emissions for high ethanol content fuels, however they did not measured the secondary aerosol for a standard low ethanol content fuel nor for ethanol fuel (E100). This study shows that SOA formation from high ethanol content is moderate to low when compared to low ethanol content fuel. Suarez-Bertoa et al. (2015) also concluded that short-chain hydrocarbons could have a role in SOA formation, not only the aromatic BTEX compounds. We observed an increase in ethanol and total hydrocarbon emissions as the ethanol content of fuel increased, however, the secondary aerosol formation was observed to be lower for these high ethanol content fuels when compared to low ethanol content fuel (E10). In this study, the concentrations of BTEX and secondary aerosol formation potential both decreased as the ethanol content of fuel increased, indicating that the BTEX compounds had a large influence on secondary aerosol formation. This conclusion is in line with results of Nordin et al. (2013), who found that light aromatic precursors (C6-C9) were responsible for 60% of formed SOA in a batch type smog chamber.

Tkacik et al., (2014) studied secondary aerosol formation from in-use vehicle emissions using a PAM chamber in a highway tunnel in Pittsburgh. Similar to our study, they observed secondary to primary PM ratios up to 10 inside the tunnel. They also found that the peak in secondary aerosol production occurred conditions equivalent to 2−3 days of atmospheric oxidation and with higher OH −oxidation values they saw a decrease in secondary aerosol formation due to continued oxidation fragmenting carbon compounds. In our experiments, the equivalent atmospheric age was approximately 3.9-6.2 hours during the CSUDC, when the most SOA formation took place. Thus, our results are likely on the lower-end compared to maximum secondary aerosol formation potential, but similar OH exposures are reached when compared SOA formation studies conducted with batch chambers (e.g. Platt et al. 2013; Gordon et al. 2014a; Nordin et al. 2013).

**3.5. Predicted SOA formation**

SOA yield is defined as (Odum et al., 1996):

$$Y = \frac{\Delta M_O}{\Delta HC},$$

(1)

Where $\Delta M_O$ is the formed secondary organic mass and $\Delta HC$ is the reacted precursor mass. Using Eq. (1), the measured VOCsand previously measured yields for these VOCs,  we can analyze why the SOA formation potential decreases as the

ethanol content in the fuel increases. We assumed that $\Delta HC$ equals the measured VOC concentration before PAM, and similarly to Platt et al., (2013), we use low-NOx yields to get an upper limit for SOA formation. The yields are listed in Table 4. For ethylbenzene, the SOA yield of m-xylene (0.38) was used (Ng et al. 2007; Platt et al. 2013). According to (Volkamer et al., 2009), acetylene ($C_2H_2$) SOA yield strongly depends on the liquid water content of aerosol. Here a value of 0.1 was assumed. The yields are corrected with corresponding wall -loss correction factors (Table 4) presented by Zhang et al. (2014).

The contribution of each measured VOC on predicted SOA is shown in supplementary (Tables S2-S4, Figs. S20-S22). According to the predictions, the decrease in the SOA formation is caused by the decrease in aromatic compounds in the exhaust when the ethanol content in the fuel is increased. The comparison between the predictions and the measurements is shown in Fig. S23. The trend in predictions generally agree with the measurements except for E100, where the predicted SOA is higher than for E85. The predicted SOA for E100 mostly comes from acetylene (Fig. S22). Thus, the measured SOA formation potential seems to depend rather on the aromatic concentrations than on the acetylene.

3.6 Temporal variation in chemical composition during the driving cycle for primary and secondary emissions

Figure 7 and supplement figures S5-S16 shows time series of organic, inorganic ions and refractory black carbon (rBC) compounds for primary emission (a) and emissions after the PAM chamber (b). The measurement setup used and the primary and secondary particulate emissions for E10 have been published previously by Karjalainen et al. (2016). Shortly, for E10, the largest PM, organic, rBC and nitrate emissions were observed at the beginning of the cycle during the first acceleration (Karjalainen et al., 2015). Occasional increases were also observed during deceleration and engine braking conditions (Rönkkö et al. 2014, Karjalainen et al. 2014). At the end of the cycle, when speed was above 70 km/h, the largest inorganic ion (sulphate, chloride and ammonium) emissions were observed. A moderate increase in organics and rBC was also observed at the end of the cycle.

Compared to E10, for which the highest primary organic emissions (peak concentration up to 25 mg m$^{-3}$) were observed right after cold start, E85 and E100 the largest primary organics (peak concentration up to E85: 1.5 mg m$^{-3}$, E100: 0.8 mg m$^{-3}$) measured either on the middle of the cycle or at the highway driving part at the end of cycle. Time trend of rBC emissions was similar for all fuels. Elevated rBC emissions were only seen at the beginning of cycle, and at the high-way driving part of the cycle (Figures S6, S10, and S16). Primary inorganic ion concentrations were highest during the highway driving part of the cycle for E10 and E85. For E100 primary inorganic ion levels were typically low, except some elevated spikes that were observed for sulphate.

Time series observed for secondary emissions was completely different when compared to primary emissions. For E10 and E85, the cold start had a dominating role in secondary aerosol formation, with a clear increase after cold start in the first part of cycle (0–390 s). Similar increase at the beginning of the cycle was not observed for E100. During the second part of the driving cycle (390–780 s), secondary organic concentrations stayed at a constant level until the end of the cycle for all fuels. In contrast, for E100, organic PM concentrations measured after the PAM chamber were stable through the cycle, with no clear maxima. We note that the speed profile and the mass concentration in Fig. 7b do not correspond to each other directly due to the broad residence time distribution of the PAM chamber (Lambe et al. 2011). Still, the figure shows that the most SOA formation is caused by the cold engine and cold after-treatment in the beginning of the cycle for both E10 and E85."

The temporal behaviour of ions during the driving cycle was very different when compared to organics. After the PAM chamber, elevated nitrate concentrations were observed at the beginning of the cycle after the cold start and at the end of the cycle during highway driving part. For E85, and E100 nitrate concentrations measured after the PAM chamber were very low (Figures S5–S16). Elevated sulphate concentrations for all fuels were measured after the PAM chamber in the middle part of the cycle and at the end during the highway driving part of the cycle (Figures S5–S16). Elevated ammonium concentrations were observed at the end of cycle during the highway driving part for all fuels. The temporal behaviour of ammonium concentration was observed to be correlated with nitrate, suggesting ammonium nitrate formation. Tkacik et al. (2014) measured high ammonium nitrate concentrations (forming from NO oxidation to $HNO_3$ with subsequent neutralization with $NH_3$) exceeding SOA concentrations by a factor of 2 in measurements conducted in a highway tunnel. In this study the average contribution of inorganic ions to submicron PM mass was always below 20 % and the contribution of ammonium nitrate was always significantly lower when compared to SOA.

### 3.7. Temporal variation in size distributions of primary and secondary PM

Number size distributions of emitted particles were measured in order to understand the changes in particulate phase when the driving conditions such as speed and engine load rapidly change. Figure 8 shows the number size distributions of primary particles for each fuel as a function of time during the driving cycle. It can be seen that for E10 fuel the emissions of particles at the size range of 25–100 nm (Dp) were far higher than for E85 and E100. The emissions of the particles at the size range of 25–100 nm depended on the driving condition, so that they existed mostly during the acceleration parts of the NEDC cycle. These particles were most likely soot mode particles consisted of black carbon. This is in line with the chemical composition results that show that, as the ethanol content of fuel increased, the rBC emissions decreased.

Figure 8 also shows that from the viewpoint of particle number the role of cold start remained important with the fuels of high ethanol content. In fact, most of the particulate emissions for E100 are related to the cold start situation. For E10, 37% of the particle number was emitted during the first part of the cycle (CSUDC, 0-391s, see Karjalainen et al., 2016). While for E85 and E100 43% and 77 % of the particle number were emitted during the CSUDC part. Although it seems that the mean particle diameter slightly decreased when the ethanol content of fuel increased, the larger soot mode particles existed in the exhaust with all fuels. However, beneficially the concentration of soot mode particles over the NEDC cycle decreased significantly when the amount of ethanol in fuel was increased. Fuel changes also clearly affected nanoparticle emissions; the emissions of nanoparticles decreased as the ethanol content of fuel increased. Still there were systematic identifiable sub-10 nm particle emissions burst with all the fuels tested possible linking the emissions of the smallest particles with lubricant oil consumption. Overall, we note that the effect of fuel was larger for soot mode particles than for nanoparticles. At the end of cycle (800–1000 s) two distinct peaks were seen for E100. Same peaks were identified in rBC time series (see Fig. S16 in Supplementary material)."

The aerosol formation after the engine cold start was also clearly seen in secondary aerosol concentrations (Fig. 9). The largest particle downstream the PAM chamber were measured about 100 s after the cycle start when enough diluted exhaust gas was accumulated in the PAM chamber. Under high pollutant concentrations, practically no sub-20 nm particles were measured downstream the PAM chamber. After around 200 seconds of the cycle, the vehicle engine and the exhaust system seemingly had warmed up, and the following particle size distributions for the rest of the cycle had similar patterns. As the fuel ethanol content increases, the size of particles during cold start as well as during the whole cycle decreases. As the ethanol content of fuel increased a clear increase in the smallest nanoparticles after the PAM chamber was observed, indicating smaller amounts of condensable vapors to grow particles inside the PAM chamber. Because nanoparticle emissions were observed to decrease as the ethanol content of fuel in primary emissions, this observation indicates that small particles can form also in the PAM

chamber e.g. via condensation on particles smaller than lower size limit of instruments used, or via nucleation. We note that Figure 9 indicates that the average particle size in exhaust emissions decreased as the ethanol content increased (also shown for average values in Figure S18). This will likely effect the efficiency of how these particles are detected with the SP-AMS since the collection efficiency of aerodynamic lens used in the SP-AMS sharply decreases in particle sizes below 30 nm. It should be taken into account that size distributions shown here are number size distributions, not mass size distributions.

## 4. Conclusions

Ethanol is used in fuels to decrease the $CO_2$ emissions of transportation and thus to reduce adverse climate effects of traffic. This study shows that the use of these fuels produces benefits by decreasing exhaust PM concentrations and thus having a positive influence on air quality. A decrease in PM was seen in both primary emissions and secondary aerosol formation potential of exhaust emitted by a modern flex-fuel DISI vehicle.

The composition of primary emissions was observed to change as the ethanol content of fuel increased. The relative contribution of rBC in particulate matter decreased, whereas the contribution of organic particulate matter and inorganic ions increased. The organics to rBC –ratios for the primary emissions of E10, E85 and E100 were 0.9, 2.1 and 3.1, respectively. For all fuels most of the particulate matter observed after the PAM chamber consisted of organic compounds (E10; 89%, E85: 79%, E100: 61%). The organics to rBC –ratios measured after the PAM chamber for E10, E85 and E100 were 12, 8.3 and 3.1, respectively. The role of cold start was observed to dominate in secondary aerosol formation for E10 and E85. For E100 no significant increase in secondary aerosol concentrations due to the cold start was. As ethanol content of fuel increased, secondary aerosol formation was observed to decrease significantly. For E10, the secondary aerosol formation was significantly larger than the primary PM emissions with secondary to primary PM –ratio of 13.4, whereas for E100 a similar increase for PM mass after the PAM chamber was not observed.

The large difference in the exhaust secondary aerosol formation between E10 and fuels with higher ethanol content can be explained by considering the emissions of potential aerosol precursors. The exhaust emissions for low-ethanol fuels contained less short-chained organic species (ethanol, formaldehyde, acetaldehyde, methane, ethene) than the exhaust for E85 and E100, but significantly more aromatic compounds (benzene, toluene, ethyl benzene, xylenes). Compounds with a low number of carbon atoms are unlikely to form secondary aerosol due to their high vapor pressure; conversely, aromatic compounds are considered the most important SOA precursors among anthropogenic hydrocarbons. Their major atmospheric sink is the reaction with the hydroxyl radical (Andino et al., 1996). It is also known that the SOA yields tend to decrease at high-$NO_x$ concentrations (Henze et al., 2008). In our case, both the $NO_x$ concentration and the aromatics concentration decreased when the fuel was changed from E10 to high-ethanol fuels, but the $NO_x$ decrease was comparatively minor. At the same time, the concentration of OH-reactive, short-chained organic species increased. These factors together cause a strong decrease in the production of aromatic hydrocarbon oxidation products, which in turn decreases the production of secondary organic aerosol. The decrease in aromatic emissions may by itself be enough to explain the SOA reduction, but one should not omit the effect of the added reactivity presented by e.g. increased ethanol emissions, which may have an inhibiting effect by taking up a larger fraction of OH (similar to the inhibition caused by isoprene in the case of biogenic SOA formation, see Kiendler-Scharr et al. (2009)). The reduction in $NO_x$ should in principle increase SOA formation, but the effect is minor compared to the inhibiting causes.

This study shows that the SP-AMS combined with the PAM chamber is an efficient tool to investigate the differences in the secondary aerosol formation potential between vehicle technologies (fuels) with a high-time resolution taking the driving conditions into account. However, the study strongly recommends including a high time-resolution particle size distribution

measurement parallel with the SP-AMS. In general, the information gathered in this study is important for legislative purposes as well as for modelers and city authorities building emission estimates.

**Acknowledgements**

Authors gratefully acknowledge support from Cluster for Energy and Environment (CLEEN Ltd) Measurement, Monitoring and Environmental Assessment (MMEA) Work package 4.5.2., the Swedish Research Council FORMAS and Annex 44 within the Advanced Motor Fuels (AMF) Agreement of the International Energy Agency's (IEA).  Sanna Saarikoski thanks the Academy of Finland for funding her work (Grant No. 259016).

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

**Figures:**

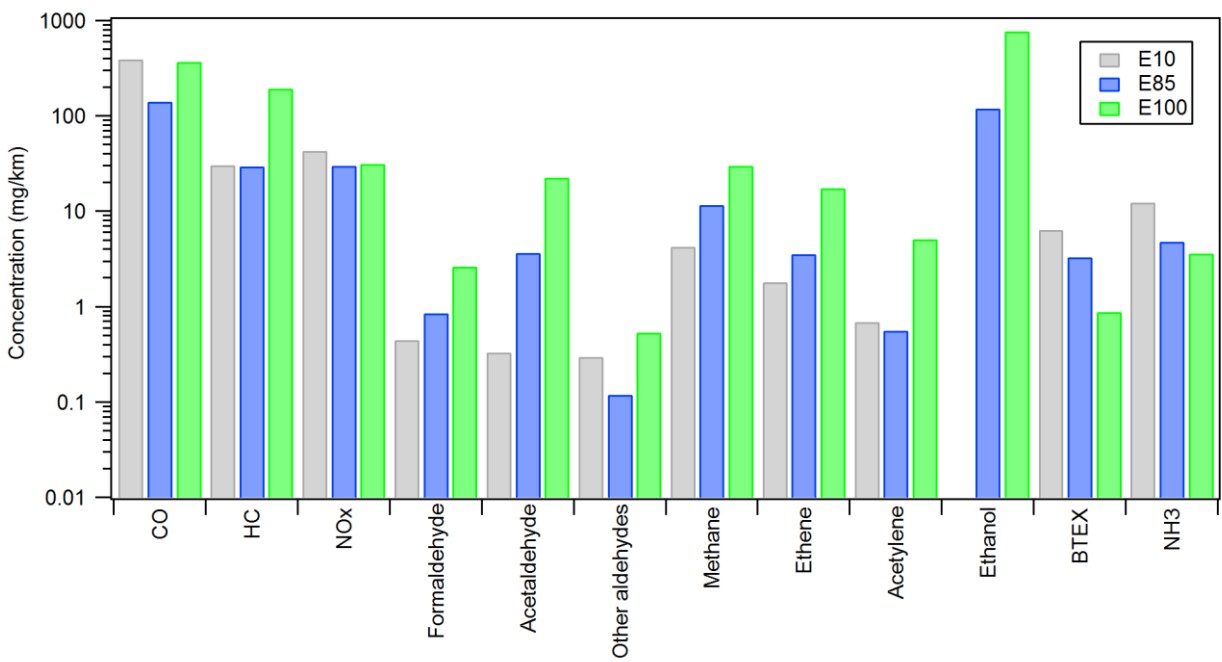

**Figure 1. Mean concentrations of gaseous compounds for different fuels measured during the NEDC cycle**

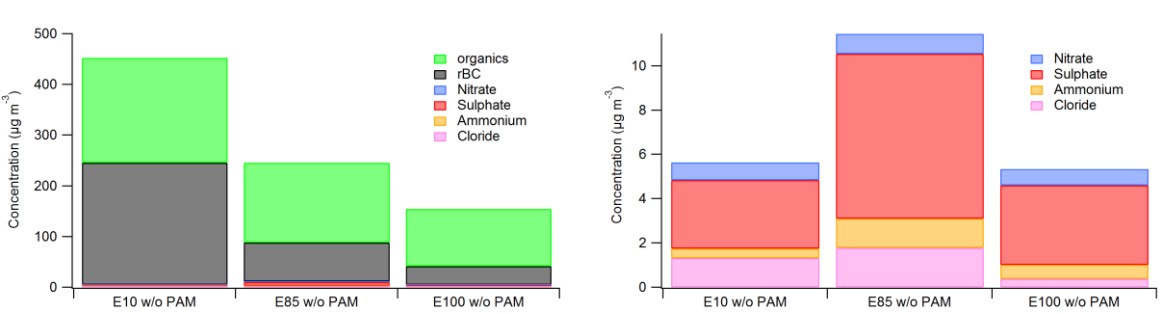

**Figure 2. Chemical composition of primary particulate emissions for E10, E85 and E100 (left). Concentrations of inorganic ions(right).**

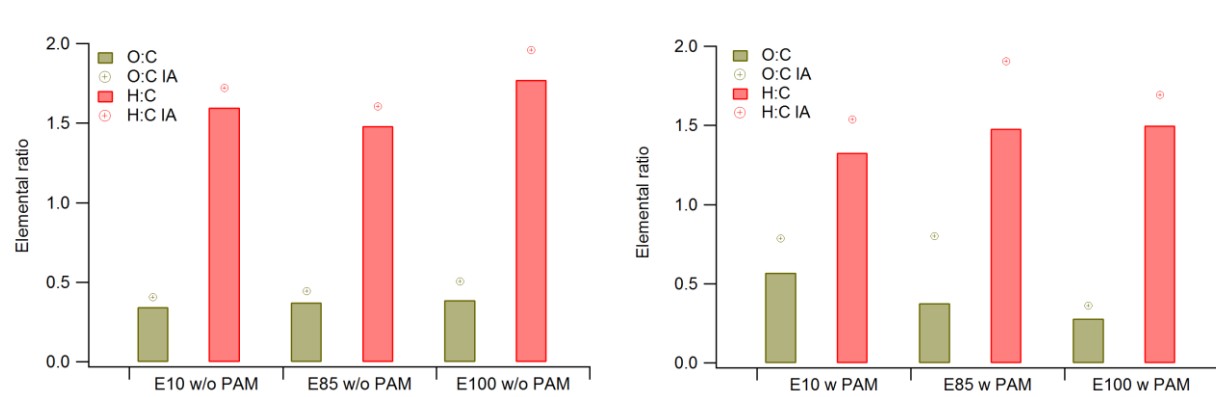

**Figure 3. Average elemental composition (O:C and H:C ratios) of emitted primary (a) and secondary (b) PM during the NEDC cycle for E10, E85 and E100. Elemental composition is calculated using original method developed by Aiken et al. (2007) and using revised method (marked with IA=improved ambient) containing more atmospherically relevant organic compounds developed by Canagaratna et al. (2015).**

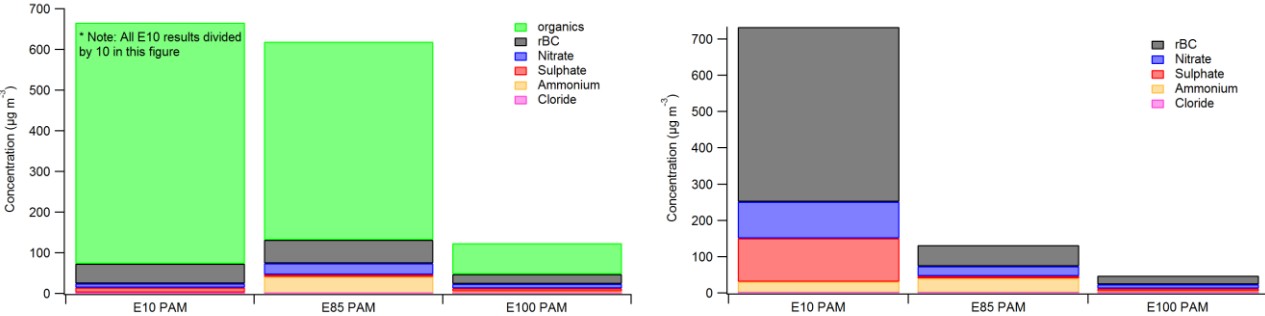

**Figure 4. Composition of particulate emissions after the PAM chamber for E10, E85 and E100 (left; Note: in left figure all E10 results are divided by 10 in order to show all results in a same y-axis). Concentration of inorganic ions and rBC after PAM chamber for E10, E85, E100 (right).**

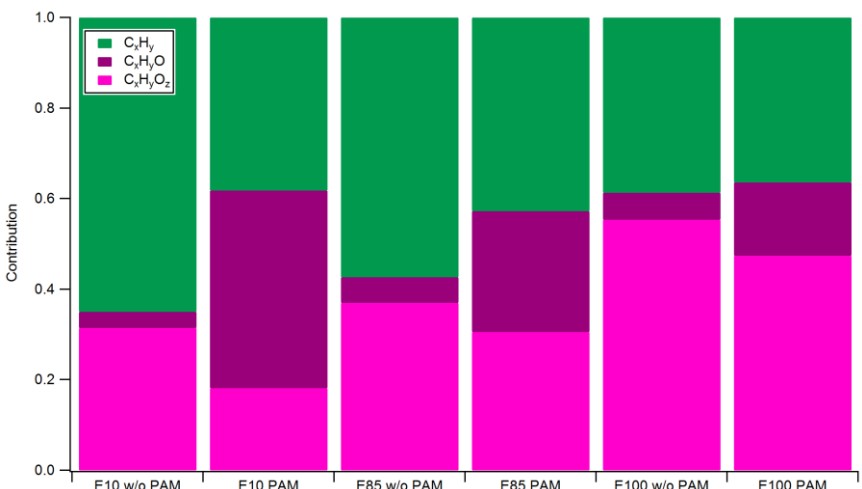

**Figure 5. Contribution of different organic fragments with and without PAM chamber. CH refers to hydrocarbons ($C_xH_y$); CHO to fragments with one oxygen atom ($C_xH_yO_z^+$) and CHO$_x$ to compounds containing several oxygen atoms ($C_xH_yO_z$, z>1).**

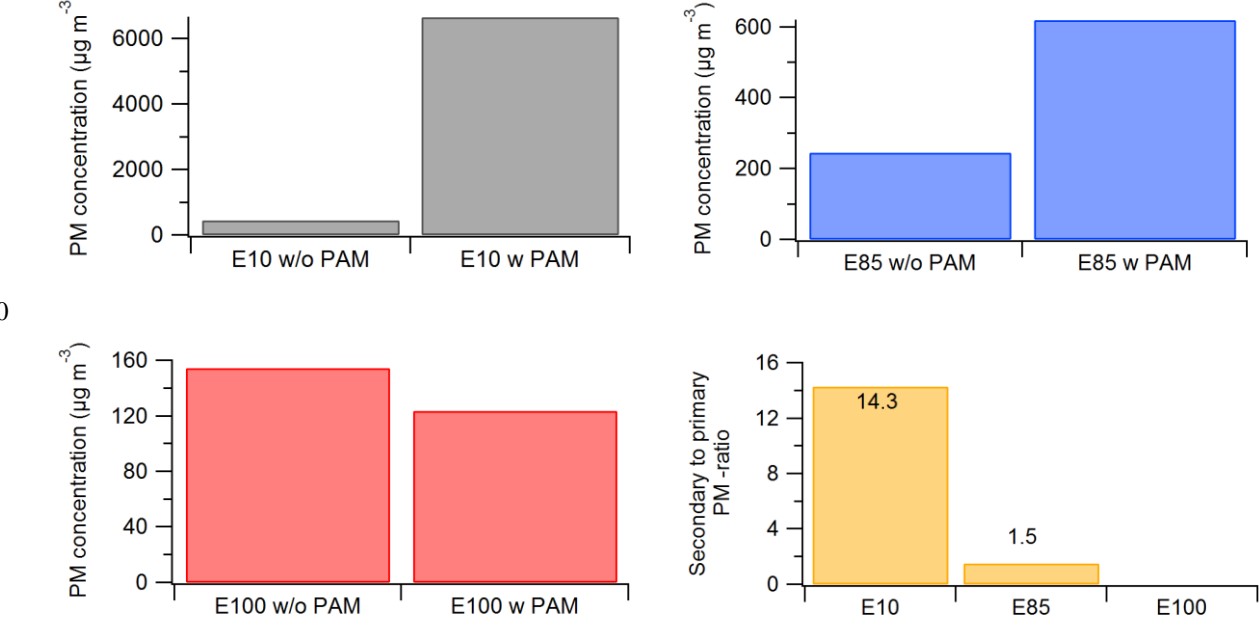

**Figure 6. Primary submicron PM concentrations in exhaust emissions measured without PAM chamber and submicron PM concentrations measured with PAM chamber for E10 (a), E85 (b) and E100 (c). The secondary to primary PM –ratio for submicron PM mass in the exhaust calculated from the SP-AMS measurements (d). Concentration of secondary particulate emission was calculated by subtracting the concentration of primary PM from the PM measured after the PAM chamber.**

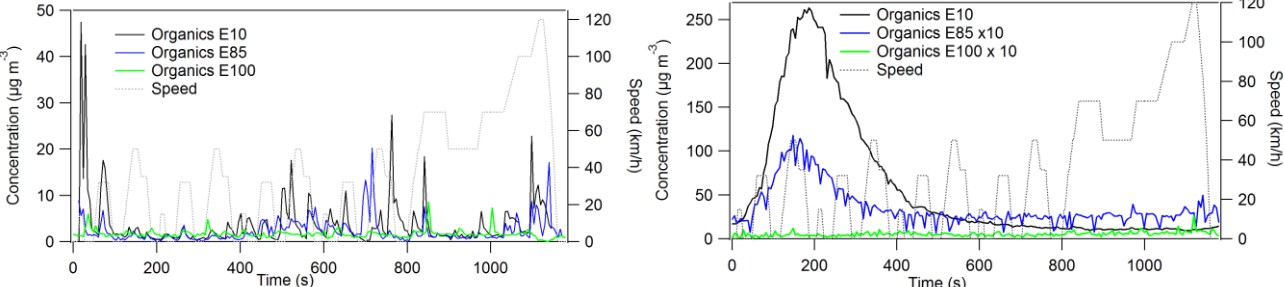

**Figure 7. Time series of organic compounds for the primary emissions (a) and for the emissions measured after the PAM chamber (b). The speed profile of the NEDC is also shown. The speed profile and the mass concentration in Fig. 7b do not correspond to each other directly due to the broad residence time distribution of the PAM chamber.**

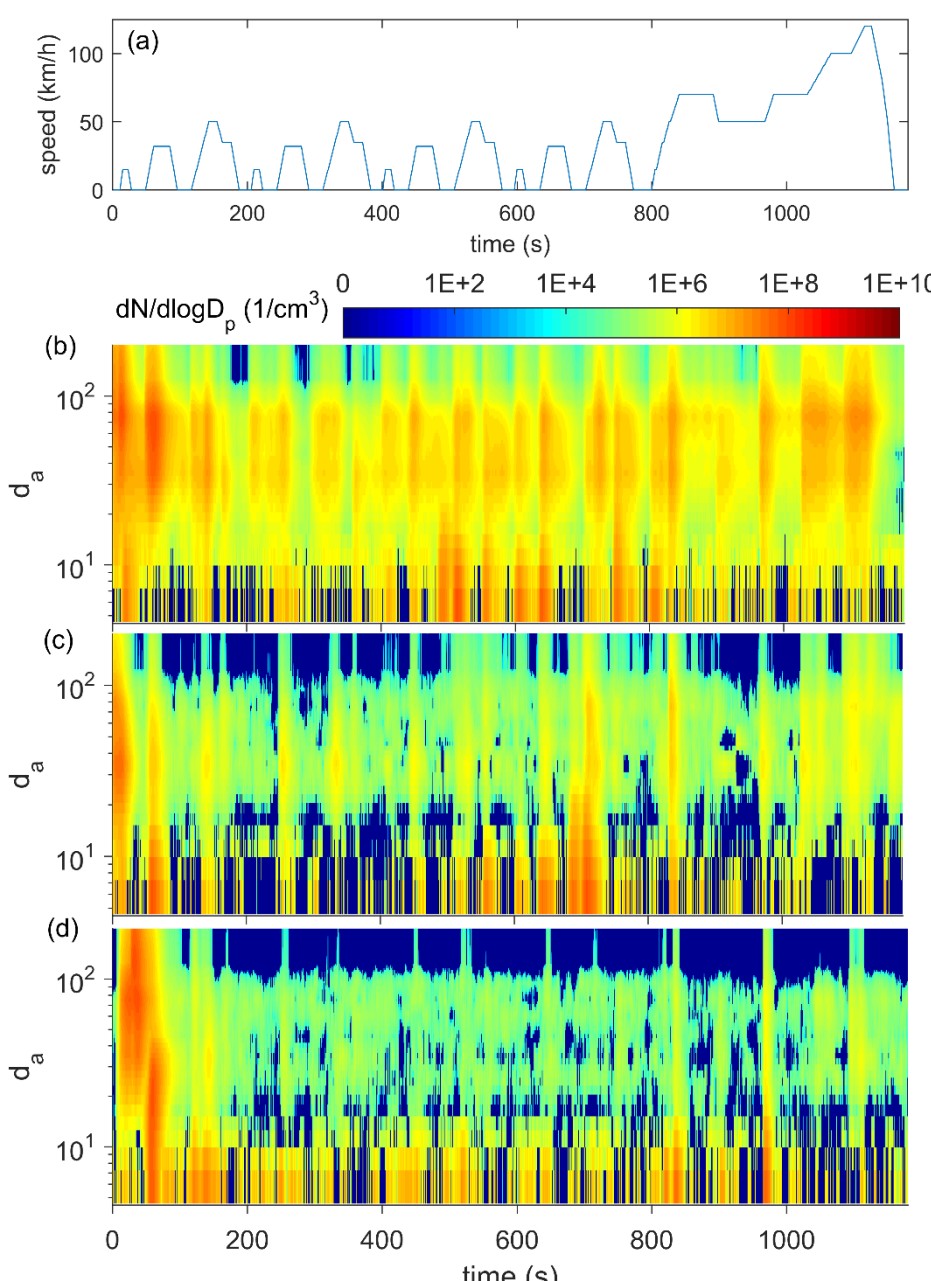

**Figure 8. Number size distributions for primary particle emissions. Measurements were made during the driving cycle for all tested fuels, E10 (b), E85 (c) and E100 (d). The driving cycle (NEDC) is shown in the panel a.**

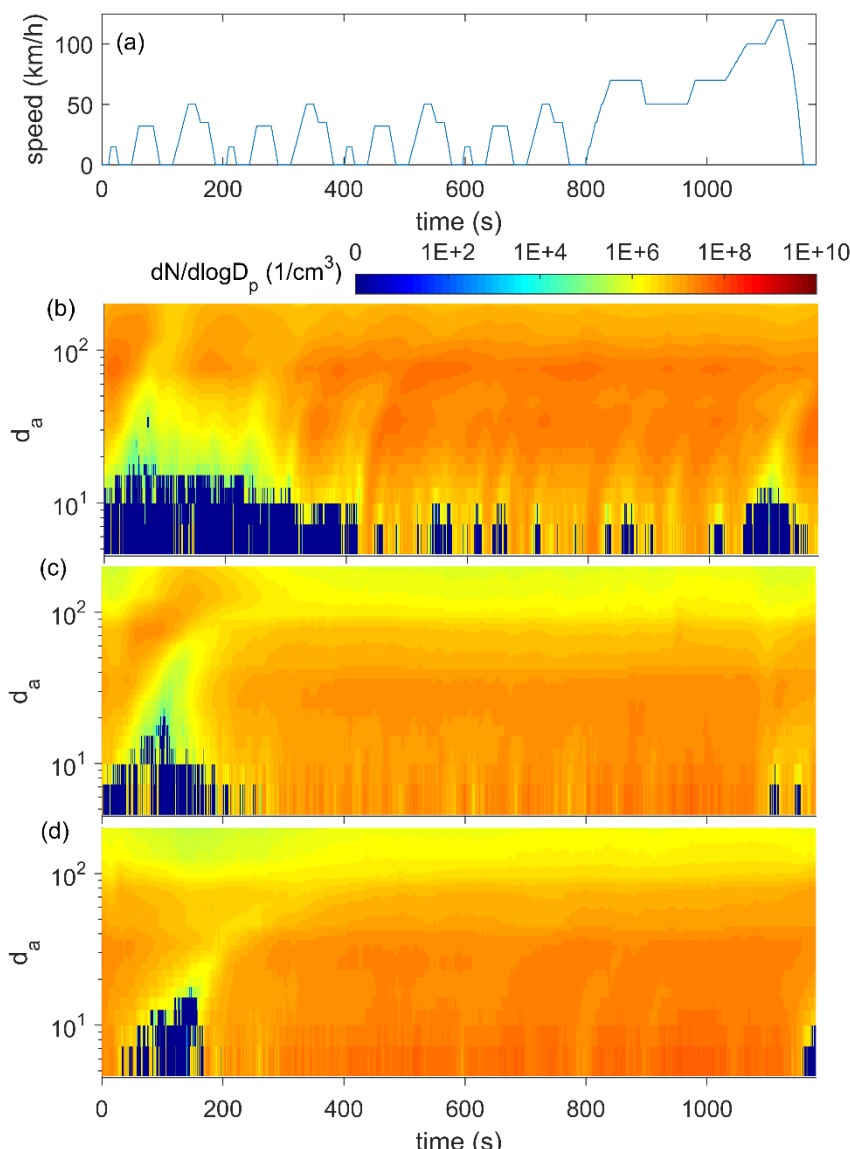

**Figure 9. Number size distributions for the secondary particle emissions measured after the exhaust treatment by the PAM. Measurements were made during the driving cycle for all tested fuels, E10 (b), E85 (c) and E100 (d). The driving cycle (NEDC) is shown in the panel (a).**

**Tables:**

**Table 1: Average OH exposures during the driving cycles**

| Fuel | CSUDC | HUDC | EUDC |
|------|-------|------|------|
| | \multicolumn | Average OH exposure (hours) | |
| E10 | 6.2 | 35.3 | 33.3 |
| E85 | 5.0 | 13.0 | 14.1 |
| E100 | 3.9 | 16.2 | 22.5 |

Table 2. Summary of measured gaseous compounds, used instruments and their detection limits.

| Instrument | Sampling | Measured compound | Detection limit |
|------------|----------|-------------------|-----------------|
| Flame ionization detector (FID) | Online | Total hydrocarbon concentration (THC) | 3ppm |
| GC (HP 5890 Series II) | Offline collection with Tedlar bag | C1-C8 hydrocarbons including methane, ethane, ethene, propane, propene, acetylene, isobutene, 1,3-butadiene, benzene, toluene, ethyl benzene and m-, p- and o-xylenes | 0.02 mol-ppm, corresponding to approximately 0.1 mg/km for methane, 0.5 mg/km for 1,3-butadiene and 0.7 mg/km for benzene. |
| HPLC, (Agilent 1260, UV detector, Nova-Pak C18 column) | Offline collection with 2,4-dinitrophenylhydrazine (DNPH) cartridges | Aldehydes; formaldehyde, acetaldehyde, acrolein, propionaldehyde, crotonaldehyde, methacrolein, butyraldehyde, benzaldehyde, valeraldehyde, m-tolualdehyde and hexanal | 0.01 mg/km. |
| Fourier transformation infrared (FTIR, Gasmet Cr-2000) | Online | $CO$, $NO$, $NO_2$, $N_2O$, Ammonia, methanol, ethanol, isobutanol, n-butanol, ETBE, formaldehyde, acetaldehyde | 2-13 ppm at 1s measurement interval corresponding to mass concentration of 1-15 mg/km over the European test cycle (Table S6)* |

**Table 3. Average concentrations of main chemical compounds of PM (in $\mu g\ m^{-3}$ and $\mu g\ km^{-1}$ (individual species) or mg $km^{-1}$(total mass)) during the NEDC cycle for primary (W/O PAM) and secondary (PAM) emissions.**

| CYCLE | ORG ($\mu g\ m^{-3}$/ $\mu g\ km^{-1}$) | NO₃ ($\mu g\ m^{-3}$/ $\mu g\ km^{-1}$) | SO₄ ($\mu g\ m^{-3}$/ $\mu g\ km^{-1}$) | NH₄ ($\mu g\ m^{-3}$/ $\mu g\ km^{-1}$) | Chl ($\mu g\ m^{-3}$/ $\mu g\ km^{-1}$) | rBC ($\mu g\ m^{-3}$/ $\mu g\ km^{-1}$) | total mass ($\mu g\ m^{-3}$/ mg $km^{-1}$) |
|-------|-----|-----|-----|-----|-----|-----|-----|
| **E10 W/O PAM** | 207.0 /158.5 | 0.8 /0.6 | 3.1 /2.4 | 0.4 /0.3 | 1.3 /1.0 | 239.9 /183.7 | 452.6 /0.346 |
| **E10 PAM** | 5928.3 /4538.9 | 101.1 /77.4 | 119.4 /91.4 | 30.6/ 23.4 | 0.9/ 0.7 | 481.3 /368.5 | 6661.7 /5.10 |
| **E85 W/O PAM** | 157.3 /122.1 | 0.9 /0.7 | 7.4 /5.8 | 1.3 /1.0 | 1.8 /1.4 | 76.7 /59.5 | 245.4 /0.190 |
| **E85 PAM** | 487.1 /378.0 | 27.2 /21.1 | 5.9 /4.6 | 40.4 /31.4 | 0.6 /0.5 | 58.0 /45.0 | 619.3 /0.48 |
| **E100 W/O PAM** | 112.7 | 0.7 | 3.6/ | 0.6 | 0.4 | 36.3 | 154.4 |

|  | /89.7 | /0.6 | 2.9 | /0.5 | /0.3 | /28.9 | /0.123 |
|---|---|---|---|---|---|---|---|
|  | 75.8 | 10.8 | 7.2 | 4.7 | 0.9 | 24.0 | 123.5 |
| **E100 W PAM** | /60.3 | /8.6 | /5.7 | /3.7 | /0.7 | 19.1 | /0.10 |

**Table 4: SOA yields for different VOCs. The vapor wall-loss correction factors are obtained from Zhang et al. (2014).**

| Compound | Yield (low-NOx) | Yield (high-NOx) | Correction (low-NOx) | Correction (high-NOx) | Reference |
|---|---|---|---|---|---|
| Toluene | 0.3 | 0.13 | 1.9 | 1.13 | Ng et al. (2007) |
| Benzene | 0.37 | 0.28 | 1.8 | 1.25 | Ng et al. (2007) |
| m/p-xylene | 0.38 | 0.08 | 1.8 | 1.2 | Ng et al. (2007) |
| 1,3-butadiene | - | 0.18 | - | - | Sato et al. (2011) |
| o-xylene | 0.1 | 0.05 | - | - | Song et al. (2007) |
| acetylene | 0.1 | - | - | - | Volkamer et al. (2009) |

