# Peer review of "Influence of fuel ethanol content on primary emissions and secondary aerosol formation potential for a modern flex-fuel gasoline vehicle"

_Atmospheric Chemistry and Physics, 2016_

## Referee Comment (RC1) · Anonymous Referee #2 · 23 Oct 2016

This study investigates the effect of fuel ethanol content on primary particulate emissions and subsequent secondary aerosol formation potential of vehicle exhaust. The authors observed a decrease in PM loadings in both primary emissions and secondary production when ethanol fuel is used during the New European Driving Cycle (NEDC) by a flex-fuel vehicle. Compared with the initial submission, the authors have significantly improved the introduction section, especially clearly defined the scope and goal of the current study. A few issues, as listed below, remain to be resolved prior to publication on ACP.

General comments

This issue has been brought out earlier yet remains elusive in the current version of the manuscript. The SOA formation potential of ethanol is significantly lower than other common anthropogenic precursors such as aromatics and along-chain alkanes considering its high vapor pressure and small molecular size. As a result, it is not surprising that substitution of regular fuel materials with equal mass of ethanol leads to a reduction in PM emissions. The key question here is essentially the energy production efficiency of the ethanol substituted fuel. The authors are suggested to normalize the reported values for primary and secondary emissions either by the total fuel mass consumed or by the total energy produced during one NEDC driving cycle. For example, a unit of microgram per cubic meter particles per gram fuel consumed would be more appropriate and illustrative to evaluate the effect of ethanol content in the fuel on the PM emissions.

The authors report the primary emissions being 0.45, 0.25, and 0.15 mg m$^{-3}$ for the E10, E85, and E100 fuel, respectively, over one driving cycle. It is necessary to give the uncertainty estimations associated with the measurements. Particularly, the authors need to measure the particle wall loss rate using inert aerosols (e.g., ammonium sulfate particles or black carbon particles) under identical experimental conditions and apply the loss rate to the measured overall primary emissions. Additionally, the authors are encouraged to discuss the uncertainties in the measured total hydrocarbon concentrations due to vapor wall losses.

The SOA yields from benzene, toluene, and xylene, and some alkanes under both low and high NOx conditions have been recently corrected by accounting for the impact of vapor wall losses (Zhang et al., PNAS, 2014). The authors are suggested to update the SOA yields values in Section 3.5, which could potentially improve the extent of agreement between predicted and measured SOA mass.

Minor comments

- Page 7, Line 25: A brief description on the gas-phase measurements of CO, $NO_x$, and ammonia needs to be given.
- Page 9, Line 13: The unit 'mg/km' is inconsistent with the unit given in Figure 2. On the other hand, the concentration unit for Figure 1 is 'mg/km'.
- Page 10, Line 23: It is interesting that the oxidized fraction of particulate organics remains the same with and without the PAM chamber. Is this because the OH-reactive small hydrocarbons consumed most of the OH radicals but did not result in organic aerosol formation due to the high volatility of oxidation products?
- Page 11, Line 23: Inaccurate statement. Small particles (e.g., $D_p < 100$ nm) exhibit high deposition rate due to molecular diffusion. But the wall deposition rate of large particles (e,g. $D_p > 300$ nm) is equally high due to the gravity induced deposition and high inertia.
- Page 16, Line 15: I am not sure if wall loss plays a role here. Are the authors indicating that the wall loss rates E100 emissions are significantly different from the E10 and E85 emissions?

---

## Referee Comment (RC2) · Anonymous Referee #3 · 25 Oct 2016

SUMMARY

In this work, Timonen et al. determine the influence of three different ethanol contents (E10, E85, and E100) on the primary emissions and secondary aerosol formation from a flex-fuel gasoline vehicle that passes EURO5 standards. Tests were conducted during a New European Driving Cycle (NEDC). While much of the work was presented as an average over the entire NEDC, the temporal variation in both chemical composition and variations in the size distributions were also explored. Aging of the exhaust was done in a Potential Aerosol Mass (PAM) chamber, which can simulate atmospheric aging on the timescales of days and with a high time resolution, unlike batch chambers. Aerosol characterization was achieved with a Soot-Photometer Aerosol Mass Spec-

Interactive
comment

trometer (SP-AMS) and an Engine Exhaust Particle Sizer; gas phase characterization was achieved with a FID, FT-IR, and GC.

In this work, the authors found that as the ethanol fuel content increased, the primary emissions of particulate matter (PM), BTEX (Benzene, Toluene, Ethylbenzene, and Xylene), the contribution of refraction black carbon (rBC) to the total PM, and the potential to form secondary PM all decreased. The concentration of total hydrocarbons as measured by FID, however, increased.

Overall, this work is sound and brings additional knowledge on the effect of fuel composition to secondary aerosol formation. There are, however, several general and specific that should be addressed increase the validity of the results and the effectiveness of the paper. Furthermore, there are many outstanding grammatical and organization errors that should also be addressed before publication.

GENERAL COMMENTS

(1) It is mentioned a couple times in this manuscript that losses through the PAM have been characterized both in general and for this system (Page 6, Line 1). The validity of this work would be greatly enhanced if these losses could be well-characterized applied to the data. This would also help interpret whether a great degree of fragmentation was causing loss of aerosol mass.

(2) How valid is it to compare the E10 and E85 PAM results to the E100 results? Looking at Figure 8, the peak aerosol production for E10 and E85 was during the CSUDC portion of the NEDC. Thus, it seems as if comparing the 1-day aging of E10/E85 to the 0.02 day aging of E100 results should either be better justified or entirely omitted from the discussion of secondary PM formation.

(3) One interesting result from this work is the chemical composition of the exhaust as function of aging. While the authors touch upon this in Figure 6, it would seem that the AMS has a much better ability to determine well-known aerosol classes through

positive matrix factorization. Is this possible with this data set? Furthermore, is this possible as a function of time during a NEDC run, or is the time resolution not sufficient? Such work could greatly enhance the impact of this paper.

SPECIFIC COMMENTS

Page 5, Line 20: Why was deionized water added to the E100 fuel to adjust water content to 4.4% m/m?

Page 6, Line 5: Why was the PAM not placed after the secondary dilution system? It seems as if that would be the most realistic in terms of the partitioning of semi-volatile organic compounds.

Page 7, Line 13: What is the reason for this difference in collection efficiencies?

Page 7, line 15: What is the significance of 90 nm for agglomerates if the restriction set by the aerodynamic lens is 50 nm?

Page 7, line 15: Furthermore, does the soot vaporizer in the SP-AMS have a lower-detection limit like the SP2? If so, that should be explicitly stated here.

Page 10, Line 24: Why are the $C_xH_yO_z$ fragments going down with PAM aging? Is this evidence of fragmentation?

Page 10, Line 30: Why are these losses not accounted for in this study?

Page 11, Line 28: The phrase "close to zero" has no physical meaning in this case. Please change to "below the detection limit of our fuel analysis" and indicate what that detection limit is.

Page 13, Line 33: If the total HC from GC, FT-IR, and HPLC, then why do the authors believe the yield is 30% lower than predicted?

Page 15, line 7-9: Why are small organic particles ruled out of this situation?

TECHNICAL COMMENTS

Page 2, Line 3: Do the authors mean to use the indefinite article "a" instead of the definite article "the" in front of New European Driving Cycle?

Page 3, line 8: The sentence starting "In addition to . . ." is very long. The clarity of this sentence could be increased by splitting it into two sentences.

Page 3, Line 9: Do the authors mean the plural "processes" instead of the singular "process?"

Page 3, Line 11: There needs to be a "," after distribution.

Page 3, Line 11: What do authors mean here by "different processes?" Are they referring to primary vs delayed primary aerosol?

Page 3, Line 13: For clarity, the authors should describe how secondary emissions differ from delayed primary. This has not been described in the manuscript yet.

Page 3, Line 18: Are there references for the previous studies using flow through chambers? That would help put this work into a better context for the reader.

Page 3, Line 20: There needs to be a "," after both instances of "chamber". Also, do the authors mean "Potential Aerosol Mass?" or "potential aerosol mass?"

Page 3, Line 23: Do the authors mean "secondary particulate emissions" or "Secondary Particulate Emissions."

Page 4, Line 1: Here, the authors have defined secondary processes, but they have mentioned it several times already. This statement would be the most effective after the first mention of secondary particulate matter.

Page 4, Line 5: The acronym FVV has not yet been defined.

Page 4, Line 3: For the sentence starting "E.g.," the use of "for example" in this sentence is redundant because E.g., is Latin for "for example."

Page 4, line 8: Since BTEX is being defined, the authors do not need to put "BTEX

=" in the parentheses. Furthermore, the right hand parentheses for this definition has been omitted.

Page 4, Line 15: It is not clear how the last two sentences of this paragraph corroborate the statement that "different photo-oxidation pathways are also dependent on conditions." I would suggest either to delete these statements or add an additional statement that clarifies why this is important.

Page 4, Line 16: Do the authors mean to add the definite article "The" in front of "European union?"

Page 4, line 23: The authors should either place a "," before typically, or move it towards the end of the sentence to read "hydrocarbon emissions are typically lower."

Page 5, Line 6: The acronym PAM has already been defined.

Page 5, Line 11: Do the authors mean "focused on" instead of "focused to?"

Page 5, line 15: "First" after the semicolon does not need to be capitalized.

Page 5, line 15: I am confused what the urban driving cycle is as opposed to the NEDC, and how it is repeated twice if doing a cold start on a separate day is required.

Page 5, line 15: This sentence is missing many definite articles. It should read "the urban," "the first," "the second," and "the last."

Page 6, Line 14: "-3" in "cm-3" should be a superscript.

Page 7, Line 1: What metals and elements specifically?

Page 7, Line 1: The phrase starting "and DeCarlo" is not an independent clause. Perhaps deleted the ";" and add a "," after "(2006)?"

Page 7, Line 31: The sentence starting "Total hydro carbons ..." is entirely redundant because an almost exact replicate of this phrase was stated on Page 7, Line 27.

Page 7, Line 34: The sentence starting "In these measurements ..." and the sentence

that follows it on Page 8, Line 1 seems entirely out of place here. First, FTIR and HPLC measurements have not been mentioned yet. This sentence makes more sense if placed after the paragraph that runs from Page 8, Line 5 to Line 10. Finally, I believe the phrase "GC and HC" should be replaced by GC, FTIR, and HPLC" for this sentence to make sense.

Page 8, Line 13: Delete "-" before "ratios."

Page 9, Line 24: Please enclose "2015" in parentheses.

Page 9, Line 26: Please enclose "2015" in parentheses.

Page 9, Line 26: "O:C ratio" is a redundant phrase as the : indicates that it is a ratio. Please remove " ratio" in this instance and all instances thereafter.

Page 12, Line 27: Do the authors mean "studied the exhaust of secondary?"

Page 13, Line 13: There should be a "," after "Thus."

Page 14, Line 4: The phrase "for E10 largest" should read "for E10, the largest."

Page 14, Line 8: The definite article "the" should come before "largest."

Page 14, Line 9: The indefinite article "A" should come before "moderate." Furthermore, the plus "were" should be changed to "was."

Page 14, Line 15: The sentence that starts "For E10 . . ." is missing commas and definite and indefinite articles. It should read "For E10, the cold start had a dominating role in secondary aerosol formation, with a clear increase after the cold start . . ."

Page 14, Line 33: What does "ions" refer to in this case? Inorganic species, ions in the mass spectrum, or both?

Page 15, Line 6: Do the authors mean "dependent" instead of "depending?"

---

## Referee Comment (RC3) · Anonymous Referee #1 · 27 Oct 2016

Timonen et al describe experiments using a flexible fuel vehicle operating on a dynamometer. They quantify the primary emissions of gases and particles, and investigate secondary PM formation using a potential aerosol mass (PAM) flow tube reactor. While the paper is topically relevant to ACP - indeed several similar papers (e.g., Chirico et al, Gordon et al, Platt et al) have been published in this journal - it is not ready for publication in its current form.

The paper is very poorly organized and presented. It reads more like a first-draft, "core dump" of all of the relevant data rather than a thoughtful analysis and discussion of the novel elements of this particular set of experiments. Furthermore, the authors have not sufficiently described the experimental setup and operation. Thus I have serious

concerns about how the data were collected and interpretation of the results that must be addressed in revision.

Specific comments:

1. Please define all abbreviations and acronyms upon first use. E.g., FFV in line 5, page 4 and THC on line 7 of the same page.

2. Page 5, Lines 15-17 - Including a time trace of the NEDC would be useful. Not all readers are familiar with the test cycle. (Upon further reading this time trace is presented in Figure 8, which is too late in the paper to be of much use to readers unless they plan to read the manuscript multiple times.)

3. How many experiments were conducted with each fuel? It seems like only one test was run per fuel, and I would argue that is not enough.

4. The methods section includes a long in-line list of measured VOCs. This information might be easier to interpret as a table that could also include information such as minimum detection limit or some of the measurement results.

5. How was the SP-AMS operated, and what data are presented? Do subsequent figures show rBC + NR-PM1 data from the laser + vaporizer configuration, or are rBC data from laser-on and the NR-PM1 data from laser-off operation? It is almost impossible for me to interpret the SP-AMS data without knowing exactly what data are being presented. Ionization efficiency from the two SP-AMS vaporization modes (thermal versus laser) is not necessarily identical (e.g., doi:10.5194/amt-2016-201), and thus it is important to know whether laser and/or thermal vaporizer data are presented.

6. Page 8, Line 18 - How was ammonia measured? It was not listed in the Methods.

7. Were any special precautions or preparations taken between experiments with different fuels? E.g., does the vehicle need to run for a certain number of kilometers on E85 after using E10 in order for the on-board oxygen sensor or other parameters to be reset? Similarly, was the vehicle certified to run on E100? The huge increase in

emissions for E100 versus E10 and E85 (Figure 1) might indicate operational problems.

8. Section 3.3.2 - rBC increased in the PAM for E10. The authors offer several possible explanations, but none seem particularly satisfying. Was there a separate measurement of BC (e.g., aethalometer) available for an independent measurement? The authors note that some of the apparent increase in rBC mass may be from growth of small particles with BC cores. While this may be the case, I would be more worried about changing particle collection and ionization efficiency in the laser-on mode (again, it would be very helpful to know how the SP-AMS was operated in this study). Did concentrations of inorganic ions (e.g., sulfate) also increase after the PAM for E10 operation? If the hypothesis that the increase in rBC comes from particle growth is correct, one might also expect to see higher sulfate from particle growth as well.

9. Section 3.4 - Wall losses in the PAM - the authors mention that wall losses may be an issue, but make no attempt to characterize or quantify them. Given that Jose Jimenez's group has recently published papers that consider PAM wall losses (e.g., doi:10.5194/acp-16-2943-2016), it seems incomplete for the authors to try and publish data without at least simple wall loss corrections.

10. Figures 2, 4, and 7 all show concentration rather than emission factor. I think plotting emission factor would be more instructive because it removes any differences in dilution between the separate experiments. In the end what is important are differences between fuels, and plotting concentration does communicate those differences, but it is hard to interpret, e.g., Fig 4, when the E10 bar is divided by 10 in the left panel but not the right. Using emission factor may help with the scaling.

11. Figure 3 and 5 would be easier to compare if the information was combined. The authors expect readers to compare O:C and H:C from fresh emissions to post-oxidation, but the figures may very well end up on different pages once typeset. It is much easier to compare adjacent bars than to look at one figure, remember the O:C, and compare it to another figure.

[Figure]

**[ACPD](ACPD)**

Interactive
comment

12. Figure 6 suggests that oxidation converts compounds that produce CxHy and CxHyO ions to compounds that produce ions with multiple oxygen. Can the authors suggest relevant chemistry that is happening? I was surprised to see that the CxHyO bar always seems to be depleted after the PAM relative to the fresh emissions. This seems like an important finding.

13. The comparison of primary versus secondary in Figure 8 is not like-versus-like. The PAM is not a perfect plug-flow apparatus. It operates more like a CSTR and there is mixing within the PAM. Thus the emissions become "smeared" over time, and this is evident from the smooth shape of the PAM output. Therefore trying to correlate the peak in PAM output to a specific point in the driving cycle seems to unwise (and perhaps impossible).

14. Likewise, the size distributions in Figure 10 likely tell you more about the gas-phase emissions entering the PAM than anything. The multiple "nucleation" events probably coincide with bursts of VOCs that can be converted to SOA.

15. In general, the discussion of the size distributions seems like a loose appendage. Section 3.7 and Figures 9 and 10 should either be expanded significantly or removed.

16. The manuscript needs a thorough edit for English grammar.

---

## Author Comment (AC1) · 31 Jan 2017

We thank the reviewers for taking time to read the article and their valuable comments on our paper. To facilitate the revision process we have copied the reviewer comments in black text. Our responses are in regular blue font. We have responded to all the reviewer comments and made alterations to our paper (**in bold text**)

**Anonymous Referee #1**

Reviewer: Timonen et al describe experiments using a flexible fuel vehicle operating on a dynamometer. They quantify the primary emissions of gases and particles, and investigate secondary PM formation using a potential aerosol mass (PAM) flow tube reactor. While the paper is topically relevant to ACP - indeed several similar papers (e.g., Chirico et al, Gordon et al, Platt et al) have been published in this journal - it is not ready for publication in its current form. The paper is very poorly organized and presented. It reads more like a first-draft, "core dump" of all of the relevant data rather than a thoughtful analysis and discussion of the novel elements of this particular set of experiments. Furthermore, the authors have not sufficiently described the experimental setup and operation. Thus I have serious concerns about how the data were collected and interpretation of the results that must be addressed in revision.

The structure of paper is clarified and more information about the measurement setup, used experimental setup (instrument, regulated emissions, cycle profile, car preparation, SP-AMS etc.) was added to manuscript and supplemental material. A detailed discussion about vapor and PM losses in the PAM chamber was added to the manuscript and supplemental material. SOA calculations and literature references are revised to include latest research results. In addition, comparison to other relevant articles is improved. The results section is revised and clarified

To our opinion, this article is now significantly improved. Article includes results from a comprehensive measurement setup containing gases and particles (primary emissions and secondary aerosol formation potential) and will provide important new information about the GDI vehicle emissions when fuels with different alcohol content are used.

Specific comments:

Please define all abbreviations and acronyms upon first use. E.g., FFV in line 5, page 4 and THC on line 7 of the same page.
All abbreviations and acronyms are carefully checked and defined when first used.

Page 5, Lines 15-17 - Including a time trace of the NEDC would be useful. Not all readers are familiar with the test cycle. (Upon further reading this time trace is presented in Figure 8, which is too late in the paper to be of much use to readers unless they plan to read the manuscript multiple times.)
Agreed. Time trace of the NEDC cycle (Fig S2) as well as following information about the NEDC cycle was added to the supplemental material.
**Driving cycle and car preparation**

Cold-start tests were carried out by using the European exhaust emissions driving cycle, "NEDC" (Fig. S2), which is defined in the UN ECE R83 regulation (Figure 1). NEDC totals 11.0 km, divided into three test phases to study emissions at cold start and with warmed-up engines. The first and second test phases each consisted of 2.026 km driving, and the third test phase, the extra-urban driving cycle (EUDC), was 6.955 km.

[Figure]

**Figure S2: NEDC driving cycle**

Preparation needs and stability issues related to the FFV cars were based in the earlier project (Aakko-Saksa et al., 2014). After the fuel change and prior to NEDC, a hot-start test was applied to monitor how warmed-up cars performed. For this purpose, the FTP75 city driving cycle was run as a hot-start test (FTP75 cold-start procedure is defined by the US Environmental Protection Agency EPA). FTP75 driving cycle totals 17.77 km, which is divided into three test phases including a 600 seconds pause. Before the FTP75 hot-start test, a "dummy" test FTP75 was conducted to stabilize cars for the actual hot-start test. Thereby, preparation of cars before the cold-start NEDC test on the following test day to avoid carry-over effect was extensive.Two NEDC tests were conducted for each fuel. Table S1 includes the concentrations of regulated emissions (± st.dev) during the driving cycle.

A reference to supplement Fig S2 was added to the manuscript "**The driving cycle was New European Driving Cycle (NEDC; cycle profile shown in Fig. S2).**

How many experiments were conducted with each fuel? It seems like only one test was run per fuel, and I would argue that is not enough.

Two NEDC tests were conducted for each fuel. While some parameters were monitored similarly during both of these (gaseous emissions, particle size distribution of primary exhaust particles), the extensive study for the differences between primary and secondary particle emissions could be conducted once per fuel. This was because the cold start was included in all of these tests causing that the total duration of study was approximately eight days. The cold start test required long conditioning time prior to the test, so that only one NEDC cycle could be conducted per day. Also the preparation of car for different fuels took extensive time. The study included also other aims (Aakko-Saksa et al., 2014) meaning that the resources did not allow us to conduct more repetitions.

We have compared the primary particle data and gaseous emission data from two separate but identically conducted NEDC cycles (Table S1). We observed that the results from these were close to each other, indicating that experiments produced

repeatable results. However, due to the limited amount of published data in this important topic, we hope that future studies produces even more data and results in order to understand e.g. the repeatability of secondary particle emission measurement. Although somewhat limited, our opinion is that our main results that fuel properties affect the primary and secondary emissions of GDI vehicle a lot and that those emissions are significantly dependent on the driving conditions are important.

Reference: Aakko-Saksa, P., Murtonen, T., Roslund, P., Koponen, P., Nuottimäki, J., Karjalainen, P., Rönkkö, T., Timonen, H., Saarikoski, S. and Hillamo, R. Research on Unregulated Pollutants Emissions of Vehicles Fuelled with Alcohol Alternative Fuels - VTT's contribution to the IEA-AMF Annex 44. Research Report : VTT-R-03970-14, 2014. VTT, 28 p. + app. 6 p.

Following clarification was added to the manuscript: **Two NEDC tests were conducted for each fuel. While some parameters were monitored similarly during both of these (gaseous emissions, particle size distribution of primary exhaust particles), the extensive study for the differences between primary and secondary particle emissions could be conducted once per fuel.**

The methods section includes a long in-line list of measured VOCs. This information might be easier to interpret as a table that could also include information such as minimum detection limit or some of the measurement results.

Agreed. A table summarizing the measured gaseous compounds, used instruments and their detection limits was added to the manuscript. In addition, to complement this a table containing FTIR detection limit for each measured compound was added to the supplement.

**Table 2**. Summary of measured gaseous compounds, used instruments and their detection limits.

| Instrument | Sampling | Measured compound | Detection limit |
|---|---|---|---|
| Flame ionization detector (FID) | Online | THC concentration | 3ppm |
| GC (HP 5890 Series II) | Offline collection with Tedlar bag | C1-C8 hydrocarbons including methane, ethane, ethene, propane, propene, acetylene, isobutene, 1,3-butadiene, benzene, toluene, ethyl benzene and m-, p- and o-xylenes | 0.02 mol-ppm, corresponding to approximately 0.1 mg/km for methane, 0.5 mg/km for 1,3-butadiene and 0.7 mg/km for benzene. |
| HPLC, (Agilent 1260, UV detector, | Offline collection with 2,4-dinitrophenylhydrazine (DNPH) cartridges | Aldehydes; formaldehyde, acetaldehyde, acrolein, propionaldehyde, crotonaldehyde, | 0.01 mg/km. |

| Nova-Pak C18 column). | | methacrolein, butyraldehyde, benzaldehyde, valeraldehyde, m-tolualdehyde and hexanal | |
|---|---|---|---|
| Fourier transformation infrared (FTIR, Gasmet Cr-2000) | Online | CO, NO, $NO_2$, $N_2O$, Ammonia, methanol, ethanol, isobutanol, n-butanol, ETBE, formaldehyde, acetaldehyde | 2-13 ppm at 1s measurement interval corresponding to mass concentration of 1-15 mg/km over the European test cycle (Table S5) |

**Table S5.** Detection limits as a ppm and mg/km for compounds measured with the FTIR

| | Detection limit | |
|---|---|---|
| | Concentration at 1-s interval (ppm) | European test (mg/km) |
| Carbon monoxide (CO) | 7 | 8 |
| Nitric oxide (NO) | 13 | 15 |
| Nitrogen dioxide (NO2) | 2/10 | 4 |
| Nitrous oxide (N2O) | 4 | 4 |
| Ammonia | 2 | 1 |
| Methanol | 2 | 1 |
| Ethanol | 4 | 7 |
| Isobutanol | 3 | 9 |
| n-Butanol | 4 | 12 |
| ETBE | 2 | 8 |
| Formaldehyde | 5 | 6 |
| Acetaldehyde | 5 | 9 |

How was the SP-AMS operated, and what data are presented? Do subsequent figures show rBC + NR-PM1 data from the laser + vaporizer configuration, or are rBC data from laser-on and the NR-PM1 data from laser-off operation? It is almost impossible for me to interpret the SP-AMS data without knowing exactly what data are being presented. Ionization efficiency from the two SP-AMS vaporization modes (thermal versus laser) is not necessarily identical (e.g., doi:10.5194/amt-2016-201), and thus it is important to know whether laser and/or thermal vaporizer data are presented.

Due to highly variable emission source with accelerations and decelerations during the NEDC driving cycle, SP-AMS was setup to measure mass spectra with 5 second time-resolution. It was not possible to switch between laser-on and off modes due to variations in measured emissions, so all data is rBC+NR-PM1 data. Also, it was not possible to record size distributions.

Following clarification about the measurement conditions was added to the text: **A five second averaging time was used in the AMS measurements and dual-vaporization system with laser and tungsten oven operating was used during the measurements. Only V-mode data is used in this study. For SP-AMS one second and 1 minute 3σ detection limits for submicrometer aerosol are <0.31 μg m-3 and < 0.03 μg m-3 for all species in the V-mode, respectively (DeCarlo et al., 2006, Onasch et al., 2012).**

Page 8, Line 18 - How was ammonia measured? It was not listed in the Methods.

Ammonia was measured with a FTIR. Information is added to the new table 2 containing instruments and measured compounds for gaseous emissions.

Were any special precautions or preparations taken between experiments with different fuels? E.g., does the vehicle need to run for a certain number of kilometers on E85 after using E10 in order for the on-board oxygen sensor or other parameters to be reset? Similarly, was the vehicle certified to run on E100? The huge increase in emissions for E100 versus E10 and E85 (Figure 1) might indicate operational problems.

Preparation needs and stability issues related to the FFV cars were known based in the earlier project (Aakko-Saksa et al., 2014). Prior the tests the FFV vehicle was conditioned according to the manufacturer's instructions, and the adaptation of the car to new fuel was monitored. Cold-start tests were carried out by using the European exhaust emissions driving cycle, "NEDC", which is defined in the UN ECE R83 regulation (Figure S2). NEDC totals 11.0 km, divided into three test phases to study emissions at cold start and with warmed-up engines. The first and second test phases each consisted of 2.026 km driving, and the third test phase, the extra-urban driving cycle (EUDC), was 6.955 km. Due to cold start requirement, NEDC tests could not be run at the fuel change day. After fuel change, hot-start test was run in order to monitor how warmed-up cars performed. For this purpose, the FTP75 city driving cycle was run as a hot-start test (FTP75 cold-start procedure is defined by the US Environmental Protection Agency EPA) several times. First a "dummy" test FTP75 was conducted to stabilize cars for the actual hot-start test and then FTP75 hot-start test(s). Thereby, preparation of cars before the cold-start NEDC test on the following test day was extensive.

Following sentences were added to the manuscript: **Preparation needs and stability issues related to the FFV cars were based in the earlier project (Aakko-Saksa et al., 2014). Shortly, the FFV vehicle was conditioned according to the manufacturer's instructions, and the adaptation of the car to new fuel was monitored.**

Aakko-Saksa, P., Rantanen-Kolehmainen, L. and Skyttä, E. Ethanol, Isobutanol, and Biohydrocarbons as Gasoline Components in Relation to Gaseous Emissions and Particulate Matter. Environmental Science & Technology. American Chemical Society. Vol. 48 ( 2014) No: 17, 10489 – 10496. doi: 10.1021/es501381h.

Section 3.3.2 - rBC increased in the PAM for E10. The authors offer several possible explanations, but none seem particularly satisfying. Was there a separate measurement of BC (e.g., aethalometer) available for an independent measurement? The authors note that some of the apparent increase in rBC mass may be from growth of small particles with BC cores. While this may be the case, I would be more worried about changing particle collection and ionization efficiency in the laser-on mode (again, it would be very helpful to know how the SP-AMS was operated in this study). Did concentrations of inorganic ions (e.g., sulfate) also increase after the PAM for E10 operation? If the hypothesis that the increase in rBC comes from particle growth is correct, one might also expect to see higher sulfate from particle growth as well.

Unfortunately it was not possible to install another BC instrument to the measurement system. The flow through PAM chamber (max 10 LPM) was limiting the amount of instruments installed after the chamber. Dual vaporizer system with laser and tungsten oven operating was used in all the measurements during this campaign. Yes, concentrations of all compounds increased after PAM. Increases for inorganic ions and organics were significantly higher (PAM/w/oPAM –ratio of 30-100) than for BC (PAM/w/oPAM –ratio of 2), for E10. However, for inorganic ions and OC this is expected due to secondary aerosol formation process. The regulated emissions (supplement table S2) show that the cycles were repeatable and soot concentration is expected to be approximately similar in both cycles. There can be several reasons, why SP-AMS detects more BC after chamber. Based on changes in size distribution (supplement figS18), it seems likely that at least following processes have influence:

1) Growth of particles due to condensation -> increased transmission in aerodynamic lens: In exhaust emissions soot mode maximum is typically in very small particles, at around 90nm. As the AMS collection efficiency due to aerodynamic lens restrictions steeply decreases for particles smaller than 50nm, growth due to condensation will likely partly increase the collection efficiency. Size distributions also show a clear increase in particle size.

2) Change in morphology due to condensation – > increased CE: Willis et al., 2014 demonstrated that also particle morphology affects the SP-AMS particle beam width, that in turn affects the collection efficiency through the overlap of the particle beam and the laser beam. This could also increase the detection efficiency of BC in this case.

However, based on this study, it is not possible estimate influence of these effects to seen increase in BC. Possibly a future study could address this issue further.

Chapter in question was clarified and following sentence was added to the manuscript:

**Based on this study, it is not possible to estimate which of these above mentioned processes is the main reason for observed BC increase.**

Section 3.4 - Wall losses in the PAM - the authors mention that wall losses may be an issue, but make no attempt to characterize or quantify them. Given that Jose Jimenez's group has recently published papers that consider PAM wall losses (e.g.,doi:10.5194/acp-16-2943-2016), it seems incomplete for the authors to try and publish data without at least simple wall loss corrections.

We agree that the estimation of wall losses and fragmentation of the low volatility vapors formed in the PAM is necessary, especially when the condensational sink is small and/or the OH exposure is high. We studied the losses using the model developed by Palm et al. (2016) and found that in all cases, more than 95 % of the LVOCs condense on particles, and thus the effect of vapor wall losses on the measured particle mass is negligible. This proportion is much higher than in previous studies (Ortega et al., 2016; Palm et al., 2016) because in our study the condensational sink is higher, which favors condensation on particles. In addition, PM losses in chamber were measured in laboratory. A chapter about PM and vapor losses in the chamber was added to the supplement and a short summary to the manuscript.

**The following test was added to the manuscript:**

The secondary aerosol in PAM chamber is formed when low volatility vapors condense on aerosols or form new particles. In the PAM chamber, these vapors may also condense onto walls, exit the chamber, or react with OH, which leads to fragmentation and increase in the saturation vapor pressure. Thus the potential aerosol mass is underestimated if these chamber related losses of low volatile vapors are not taken into account. We used the LVOC (low volatility organic compound) fate model presented by Palm et al. (2016) to estimate the losses of condensing organic vapors in the PAM chamber.

PM losses in the chamber were studied in laboratory using similar PAM chamber as in the measurements. Supplemental material includes a detailed description of loss calculations and measured PM losses as a function of particle size. Shortly, losses of primary PM in a PAM chamber (Fig. S3) are in general small especially in the particle sizes that contain most of the aerosol mass: 25% at 50 nm, 15% at 100nm and below 10% above 150 nm. Also, because of the high condensational sink, over 95 % of the LVOCs condensed on aerosol in all cases according to this model. Thus, the chamber related losses of LVOCs and PM are small.

**The following section was added to Supplementary:**

The secondary aerosol is formed when low volatility vapors condense on aerosols or form new particles. In the PAM chamber, these vapors may also condense onto walls, exit the chamber, or react with OH, which leads to fragmentation and increase in the saturation vapor pressure. Thus the potential aerosol mass is underestimated if these chamber related losses of low volatile vapors are not taken into account. We use the LVOC (low volatility organic compound) fate model presented by Palm et al. (2016) to estimate the losses of condensing organic vapors in the PAM chamber.

In the model, the relative fates of the vapor are estimated by studying the timescales of condensation on particles, condensation on chamber walls, reaction with OH radical and the residence time in the PAM chamber. Detailed description of the model can be found in Palm et al. (2016). Shortly: The lifetime for LVOC condensation is

$$\tau_{aer} = \frac{1}{4\pi \cdot CS \cdot D},$$

where D is the diffusion coefficient of the condensing molecule and CS is the condensational sink, which is calculated using the average of the HRLPI size distributions before and after PAM. The rate of LVOC loss to the walls is

$$\frac{1}{\tau_{wall}} = \frac{A}{V}\frac{2}{\pi}\sqrt{k_e D},$$

where A/V is the surface-area-to-volume ratio of the chamber, $k_e$ is the coefficient of eddy diffusion and D the diffusion coefficient.

The assumptions used in the model are same as in Palm et al. (2016): $D = 7 \times 10^{-6} \ m^2 s^{-1}$, $\alpha = 1$, mean free path $\lambda_g = 3\sqrt{\frac{\pi m_g}{8kT}}D \approx 1.173 \times 10^{-7} \ m$ (Pirjola et al. 1999), $\frac{A}{V} = 25 \ m^{-1}$ and $k_e = 0.0036 \ s^{-1}$.

The reaction rate constant for the reaction with OH is $k_{OH} = 1 \times 10^{-11} \ cm^3 \ molec.^{-1} \ s^{-1}$. LVOC is considered to fragment and form high-volatility molecules after five reactions with OH radical. Thus, the lifetime for fragmentation is

$$\tau_{OH} = \frac{5}{k_{OH} \cdot [OH]},$$

where [OH] is calculated based on the OH exposure and the residence time.

Using these lifetimes, the fate of LVOCs was calculated for each fuel and each part of the driving cycle, and the results are presented in Table S3. Because of the high condensational sink, over 95 % of the LVOCs condensed on aerosol in all cases according to this model. Thus, the chamber related losses of LVOCs are small.

**PM losses**

PAM chamber was designed with lower surface-area-to-volume (SA/V) ratio to minimize wall effects. Primary particle losses were measured in laboratory for a similar PAM chamber (Fig. S3) as was used in this study. Losses were in general quite small in the particle sizes that contains most of the aerosol mass: 25% at 50 nm, 15% at 100 nm and below 10% above 150 nm.

[Figure]

**Figure S3. Primary particle losses in a similar PAM chamber that was used in the study. (Figure adapted from Karjalainen et al., 2016 with permission).**

Figures 2, 4, and 7 all show concentration rather than emission factor. I think plotting emission factor would be more instructive because it removes any differences in dilution between the separate experiments. In the end what is important are differences between fuels, and plotting concentration does communicate those differences, but it is hard to interpret, e.g., Fig 4, when the E10 bar is divided by 10 in the left panel but not the right. Using emission factor may help with the scaling.

Yes, oxygen containing ethanol has lower energy content than the hydrocarbon fuels leading to different exhaust volumes per km driven. As a result, concentrations in the exhaust are not directly comparable for ethanol and hydrocarbon fuels. When concentrations are converted to mass per km unit, differences in exhaust volumes are compensated. The results are presented as mass per km in supplement (Figure S4: The composition and concentration (mg/km) of emitted primary (a) and secondary (b) PM for each fuel). This unit takes into account different energy contents of fuels.

Following text was added to the supplement prior to figure S4.

**"Oxygen containing ethanol has lower energy content than the hydrocarbon fuels leading to different exhaust volumes per km driven. As a result, concentrations in the exhaust are not directly comparable for ethanol and hydrocarbon fuels. When concentrations are converted to mass per km unit, differences in exhaust volumes are compensated."**

Figure 3 and 5 would be easier to compare if the information was combined. The authors expect readers to compare O:C and H:C from fresh emissions to postoxidation, but the figures may very well end up on different pages once typeset. It is much easier to compare adjacent bars than to look at one figure, remember the O:C, and compare it to another figure.

Agreed. The figure 5 is moved next to the figure 3 in order to enable comparison.

Figure 6 suggests that oxidation converts compounds that produce $CxHy$ and $CxHyO$ ions to compounds that produce ions with multiple oxygen. Can the authors suggest relevant chemistry that is happening? I was surprised to see that the $CxHyO$ bar always seems to be depleted after the PAM relative to the fresh emissions. This seems like an important finding.

Figure 6 represents the contribution of organic fragments with different amount of oxygen to organic fraction. For E10, E85 the absolute concentration of each organic group increases, although the contribution of $CxHyO$ slightly decreases. Also, mass spectra shows that in compound level differences for primary and secondary emissions can be observed. For E100, both contributions and concentrations of different organic families are similar with and without PAM chamber, however again a change in composition of these hydrocarbon groups is observed. Also, one must note that the sum of oxidated hydrocarbons increases in the chamber when compared to hydrocarbons.

Following clarification was added to end of this chapter:

**For all fuels, the total contribution of oxidated compounds (sum of $CxHyO$, $CxHyOz$) increased when compared to contribution of hydrocarbons. Also, for E10, E85 the mass concentration organic fraction and also concentration of**

**each organic hydrocarbon group increases in PAM chamber, although the contribution of CxHyO slightly decreased. Also, mass spectra (Fig. S11-S22) shows that in hydrocarbon composition after PAM chamber clear differences can be observed. For E100, both contributions and concentrations of different hydrocarbon families are similar with and without PAM chamber, however again a change in the composition of these hydrocarbon groups was observed.**

The comparison of primary versus secondary in Figure 8 is not like-versus-like. The PAM is not a perfect plug-flow apparatus. It operates more like a CSTR and there is mixing within the PAM. Thus the emissions become "smeared" over time, and this is evident from the smooth shape of the PAM output. Therefore trying to correlate the peak in PAM output to a specific point in the driving cycle seems to unwise (and perhaps impossible).

We agree that the broad residence time distribution of the PAM does not allow for high time-resolution analysis of the secondary aerosol formation. However, it is evident in Fig. 8 that the most SOA formation is caused by the cold engine and cold after-treatment in the beginning of the cycle for both E10 and E85, and therefore we think the figure is useful. We modified the section referring to Fig. 8 as following:

"Time series observed for secondary emissions was completely different when compared to primary emissions. For E10 and E85 cold start had dominating role in secondary aerosol formation with a clear increase after cold start in the first part of cycle (0–390 s). Similar increase at the beginning of the cycle was not observed for E100. During the second part of the driving 20 cycle (390–780 s), secondary organic concentrations level stayed at a constant level until end of the cycle for all fuels. In contrast, for E100, organic PM concentrations measured after the PAM chamber were stable through the cycle, with no clear maxima. **We note that the speed profile and the mass concentration in Fig. 8b do not correspond to each other directly due to the broad residence time distribution of the PAM chamber (Lambe et al. 2011). Still, the figure shows that the most SOA formation is caused by the cold engine and cold after-treatment in the beginning of the cycle for both E10 and E85.**"

And the figure title as following:

"Figure 8. Time series of organic compounds for the primary emissions (a) and for the emissions measured after the PAM chamber (b). The speed profile of the NEDC is also shown. **The speed profile and the mass concentration in Fig. 8b do not correspond to each other directly due to the broad residence time distribution of the PAM chamber.**"

Likewise, the size distributions in Figure 10 likely tell you more about the gas-phase emissions entering the PAM than anything. The multiple "nucleation" events probably coincide with bursts of VOCs that can be converted to SOA.

Yes, the particle growth inside the PAM chamber is caused by gases, as mentioned, most likely organic compounds. This way the particle output of the PAM chamber is defined by gas-phase phase tailpipe emissions, but especially here the focus is on gases which vapor pressure becomes lower after partial oxidation process. It is unsure whether the particle formation and growth events are due to nucleation in the PAM chamber or whether the emissions of very small particles occur during under some driving conditions. We think the latter is true. It is most likely that these were actually delayed primary particle emissions, and these particles later grew in the PAM chamber by condensation. The simultaneous emission of very small nanoparticles and VOCs can especially occur during engine motoring events (Rönkkö et al., 2014; Karjalainen et al., 2014; Karjalainen et al., 2016b). Discussion about this is added to the manuscript to section 3.7. See the next answer.

In general, the discussion of the size distributions seems like a loose appendage. Section 3.7 and Figures 9 and 10 should either be expanded significantly or removed.

We decided to expand the discussion around Figures 9 and 10 because they have an important role of showing the story of size-distribution evolution of delayed primary and secondary particles over the test cycle. The size distribution also indicate which particles are potentially detected by the SP-AMS analysis and which are left out from the analysis.

**Figure 9 related text was extended as follows.**

[revised manuscript text omitted]

The manuscript needs a thorough edit for English grammar.

The English grammar is thoroughly reviewed by a native english speaker.

**Anonymous Referee #2**

Reviewer: This study investigates the effect of fuel ethanol content on primary particulate emissions and subsequent secondary aerosol formation potential of vehicle exhaust. The authors observed a decrease in PM loadings in both primary emissions and secondary production when ethanol fuel is used during the New European Driving Cycle (NEDC) by a flex-fuel vehicle. Compared with the initial submission, the authors have significantly improved the introduction section, especially clearly defined the scope and goal of the current study. A few issues, as listed below, remain to be resolved prior to publication on ACP.

General comments

This issue has been brought out earlier yet remains elusive in the current version of the manuscript. The SOA formation potential of ethanol is significantly lower than other common anthropogenic precursors such as aromatics and along-chain alkanes considering its high vapor pressure and small molecular size. As a result, it is not surprising that substitution of regular fuel materials with equal mass of ethanol leads to a reduction in PM emissions. The key question here is essentially the energy production efficiency of the ethanol substituted fuel. The authors are suggested to normalize the reported values for primary and secondary emissions either by the total fuel mass consumed or by the total energy produced during one NEDC driving cycle. For example, a unit of microgram per cubic meter particles per gram fuel consumed would be more appropriate and illustrative to evaluate the effect of ethanol content in the fuel on the PM emissions.

Yes, the energy content of ethanol is lower. The results are presented as mass per km in supplemental material. This unit takes into account different energy contents of fuels. Oxygen containing ethanol has lower energy content than the hydrocarbon fuels leading to different exhaust volumes per km driven. As a result, concentrations in the exhaust are not directly comparable for ethanol and hydrocarbon fuels. When concentrations are converted to mass per km unit, differences in exhaust volumes are compensated.

The authors report the primary emissions being 0.45, 0.25, and 0.15 mg m-3  for the E10, E85, and E100 fuel, respectively, over one driving cycle. It is necessary to give the uncertainty estimations associated with the measurements. Particularly, the authors need to measure the particle wall loss rate using inert aerosols (e.g., ammonium sulfate particles or black carbon particles) under identical experimental conditions and apply the loss rate to the measured overall primary emissions. Additionally, the authors are encouraged to discuss the uncertainties in the measured total hydrocarbon concentrations due to vapor wall losses.

Two NEDC tests were conducted for each fuel. While some parameters were monitored similarly during both of these (gaseous emissions, particle size distribution of primary exhaust particles), the extensive study for the differences between primary and secondary particle emissions could only be conducted once per fuel. However, based on measured regulated emission measurements (table S2), the cycle was repeatable and results between cycles should be well comparable. Also, we measured PM losses in PAM chamber in laboratory and used Palm et al., 2016 model to estimate the vapor wall losses in the PAM chamber. See the answer for reviewer #1 for detailed information about the vapor and PM losses in the chamber. A summary of losses was added to the manuscript and a chapter to supplement.

The SOA yields from benzene, toluene, and xylene, and some alkanes under both low and high NOx conditions have been recently corrected by accounting for the impact of vapor wall losses (Zhang et al., PNAS, 2014). The authors are suggested to update the SOA yields values in Section 3.5, which could potentially improve the extent of agreement between predicted and measured SOA mass.

We present all the yields and the correction factors obtained from Zhang et al. (2014) now in Table 4, and the predicted SOA mass is now calculated using these corrected yields. This improvement affects also Tables S2-S4, and we also added the SOA predictions using high-NOx yields to these tables. The CO reaction rate constant is also corrected from 1.03e-13 to 2.37e-13, which changes the OHR calculations a little.

The Predicted SOA formation section is re-written using the corrected yields as follows:

"**The yields are listed in Table 4. For ethylbenzene, the SOA yield of m-xylene (0.38) was used (Ng et al. 2007; Platt et al. 2013). According to (Volkamer et al., 2009), acetylene ($C_2H_2$) SOA yield strongly depends on the liquid water content of aerosol. Here a value of 0.1 was assumed. The yields are corrected with corresponding wall-loss correction factors (Table 4) presented by Zhang et al. (2014). The contribution of each measured VOC on predicted SOA is shown in supplementary (Tables S2-S4, Figs. S5-S7). For E10, the predicted SOA mass was 5780 μg m$^{-3}$ (4.08 mg km$^{-1}$), for E85 800 μg m$^{-3}$ (0.59 mg km$^{-1}$) and for E100 1281 μg m$^{-3}$ (0.94 mg km$^{-1}$). The measured SOA for E10 was approximately the same as the predicted maximum SOA. In contrast, for E85 the measured SOA is 60 % lower than the predicted SOA, and for E100 no SOA formation was observed even though there are SOA precursors present in the exhaust gas.**

**The discrepancy between the predicted and measured SOA may result from the presence of $NO_x$ in the exhaust. The predicted SOA is an upper limit for SOA formation, based on the low-$NO_x$ yields, which are higher than the high-$NO_x$ yields (Table 4). Using the wall-loss corrected high-$NO_x$ yields, the predicted SOA emission factors are 1.37 mg**

**km$^{-1}$, 0.25 mg km$^{-1}$ and 0.53 mg km$^{-1}$ for E10, E85 and E100, respectively. Thus, the measured SOA for E10 is approximately 2.9 times higher than the predicted SOA using high-NO$_x$ yields, indicating that there are other VOCs or IVOCs contributing to SOA formation than the measured ones, or that the NO$_x$ chemistry in PAM chamber is different than that of the smog chambers where the high-NO$_x$ yields are measured. For E85, the predicted SOA using high-NOx yields is approximately the same as the measured SOA."**

**Table 4: SOA yields for different VOCs. The vapor wall-loss correction factors are obtained from Zhang et al. (2014).**

| Compound | Yield (low-NOx) | Yield (high-NOx) | Correction (low-NOx) | Correction (high-NOx) | Reference |
|---|---|---|---|---|---|
| Toluene | 0.3 | 0.13 | 1.9 | 1.13 | Ng et al. (2007) |
| Benzene | 0.37 | 0.28 | 1.8 | 1.25 | Ng et al. (2007) |
| m/p-xylene | 0.38 | 0.08 | 1.8 | 1.2 | Ng et al. (2007) |
| 1,3-butadiene | - | 0.18 | - | - | Sato et al. (2011) |
| o-xylene | 0.1 | 0.05 | - | - | Song et al. (2007) |
| acetylene | 0.1 | - | - | - | Volkamer et al. (2009) |

Minor comments

- Page 7, Line 25: A brief description on the gas-phase measurements of CO, NOx, and ammonia needs to be given.
  A new table (table 2) summarizing the gas phase measurements, instruments and corresponding detection limits was added to chapter 2.3. Gaseous phase composition measurements.

- Page 9, Line 13: The unit 'mg/km' is inconsistent with the unit given in Figure 2. On the other hand, the concentration unit for Figure 1 is 'mg/km'.
  The sentence was badly formulated. **Correction was added to the manuscript: The chemical composition of primary particulate emissions was observed to vary for different fuels (Fig. 2). Concentration for each chemical component in units mg/km for both primary and secondary emissions is shown in the supplement Fig S1.**

- Page 10, Line 23: It is interesting that the oxidized fraction of particulate organics remains the same with and without the PAM chamber. Is this because the OH- reactive small hydrocarbons consumed most of the OH radicals but did not result in organic aerosol formation due to the high volatility of oxidation products?

Yes, it is possible. Also has to be noticed that these are relative contributions. For E10 and E85 the mass concentration is for OC is significantly increased in PAM chamber and thus the concentration can be higher even though the contribution is lower. Also, one must note that the sum of oxidated hydrocarbons increases in the chamber when compared to hydrocarbons.

Following clarification was added to end of this chapter:

**For all fuels, the total contribution of oxidated compounds (CxHyO, CxHyOz) increased when compared to contribution of hydrocarbons. However, for E10, E85 the mass concentration organic fraction and also concentration of each organic hydrocarbon group increases in PAM chamber, although the contribution of CxHyO slightly decreased. Also, mass spectra (Fig. S11-S22) shows that in hydrocarbon composition clear differences can be observed. For E100, both contributions and concentrations of different organic families are similar with and without PAM chamber, however again a change in composition of these hydrocarbon groups is observed.**

- Page 11, Line 23: Inaccurate statement. Small particles (e.g., $D_p < 100$ nm) exhibit high deposition rate due to molecular diffusion. But the wall deposition rate of large particles (e,g. $D_p > 300$ nm) is equally high due to the gravity induced deposition and high inertia

  Yes, correct. However, this sentence was referring to measurement results published by Karjalainen et al., 2015, where losses of different size particles in the similar PAM chamber were measured with a CPC. There was observed that the losses were considerably higher for small particles. The sentences is reformulated:

  **We also note that the losses for PM in the PAM chamber are depending on particle size, the smaller the particles are, the larger the losses in PM were measured to be by Karjalainen et al., 2015 (Figure S3 in the supplemental material).**

- Page 16, Line 15: I am not sure if wall loss plays a role here. Are the authors indicating that the wall loss rates E100 emissions are significantly different from the E10 and E85 emissions?

  This sentence was badly formulated and is removed from the final manuscript.

**Anonymous Referee #3**

SUMMARY

In this work, Timonen et al. determine the influence of three different ethanol contents (E10, E85, and E100) on the primary emissions and secondary aerosol formation from a flex-fuel gasoline vehicle that passes EURO5 standards. Tests were conducted during a New European Driving Cycle (NEDC). While much of the work was presented as an average over the entire NEDC, the temporal variation in both chemical composition and variations in the size distributions were also explored. Aging of the exhaust was done in a Potential Aerosol Mass (PAM) chamber, which can simulate atmospheric aging on the timescales of days and with a high time resolution, unlike batch chambers. Aerosol characterization was achieved with a Soot-Photometer Aerosol Mass Spectrometer (SP-AMS) and an Engine Exhaust Particle Sizer; gas phase characterization was achieved with a FID, FT-IR, and GC.

In this work, the authors found that as the ethanol fuel content increased, the primary emissions of particulate matter (PM), BTEX (Benzene, Toluene, Ethylbenzene, and Xylene), the contribution of refraction black carbon (rBC) to the total PM, and the potentialto form secondary PM all decreased. The concentration of total hydrocarbons as measured by FID, however, increased. Overall, this work is sound and brings additional knowledge on the effect of fuel composition to secondary aerosol formation. There are, however, several general and specific that should be addressed increase the validity of the results and the effectiveness of the paper. Furthermore, there are many outstanding grammatical and organization errors that should also be addressed before publication.

GENERAL COMMENTS

It is mentioned a couple times in this manuscript that losses through the PAM have been characterized both in general and for this system (Page 6, Line 1). The validity of this work would be greatly enhanced if these losses could be well-characterized applied to the data. This would also help interpret whether a great degree of fragmentation was causing loss of aerosol mass. We agree that the estimation of wall losses and fragmentation of the low volatility vapors formed in the PAM is necessary, especially when the condensational sink is small and/or the OH exposure is high. We studied the losses using the model developed by Palm et al. (2016) and found that in all cases, more than 95 % of the LVOCs condense on particles, and thus the effect of vapor wall losses on the measured particle mass is negligible. See the answer for reviewer #1.

How valid is it to compare the E10 and E85 PAM results to the E100 results? Looking at Figure 8, the peak aerosol production for E10 and E85 was during the CSUDC portion of the NEDC. Thus, it seems as if comparing the 1-day aging of E10/E85 to the 0.02 day aging of E100 results should either be better justified or entirely omitted from the discussion of secondary PM formation.

 We significantly improved the evaluation of OH exposure by using a photochemical model, where we included the NOx emissions, which were omitted in the previous evaluation. In addition, we found that the ethanol concentration for E100 exhaust presented in Table S4 was not corrected with the dilution factor. Because of this mistake, the OHR in E100 exhaust was an order of magnitude higher than that of the other fuels, and thus the aging of E100 exhaust was under-estimated. Using the photochemical model, we got the following average equivalent atmospheric ages during CSUDC: 6.2 hours for E10, 5.0 hours for E85 and 3.9 hours for E100. We think that these ages are well comparable, and they are also the same level of oxidation as in batch chamber experiments (e.g. Platt et al. 2013, Gordon et al. 2014a, Nordin et al. 2013). The ethanol concentration and OHR in Table S4 is now corrected.

The PAM OH exposure section is re-written as follows:
**"The PAM chamber was calibrated following the procedure described by Lambe et al. (2011). According to this off-line calibration, the upper limit average OH exposure during the experiments was $1.0 \times 10^{12}$ molec. cm$^{-3}$ s, corresponding to atmospheric aging of 8 days (assuming an average OH concentration of $1.5 \times 10^{6}$ molec. cm-3 in the atmosphere (Mao et al., 2009)), but the real OH exposure is lower due to high concentrations of OH reactive gases in the exhaust. This effect is significant especially at the beginning of the cycle (CSUDC) when the concentrations are high. The average external OH reactivities (OHR) due to VOCs and CO in CSUDC were 1246 s$^{-1}$, 1141 s$^{-1}$ and 3441 s$^{-}$**

**[1] for E10, E85 and E100, respectively. The higher OHR of E100 is due to high ethanol and aldehyde concentrations. After the CSUDC, the average OHR is below 90 $s^{-1}$ for all fuelsMore detailed OHR calculations are shown in Tables S2-S4.**

**We estimate the OH exposure in the PAM by using a simple photochemical box model made by William Brune, in which the differential equations describing the chemical reactions are solved using Euler's method. More details and the source code of the model are found in PAM users manual (https://sites.google.com/site/pamusersmanual/7-pam-photochemistry-model). The free parameters in the model are photon fluxes at 254 nm and 185 nm wavelength. Based on the off-line calibration, the best-fit values for the photon fluxes are photons $cm^{-2}$ $s^{-1}$ and photons $cm^{-2}$ $s^{-1}$ for 254 nm wavelength and 185 nm wavelength, respectively. The inputs for the model are OHR due to VOCs, CO concentration, NO concentration and $NO_2$ concentration. In the model, $SO_2$ is used as a proxy for VOCs, i.e,. in the model, the OHR of $SO_2$ equals the input OHR due to VOCs. This method is reasoned to be a realistic approximation by Peng et al. (2015) in terms of estimating the OH exposure.**

**The input values for the model are obtained from 1-second time resolution measurements of CO, $NO_x$ and total hydrocarbons (THC), corrected with the residence time distribution caused by the PAM chamber. The residence time distribution is obtained from the $CO_2$ pulse experiment presented by Lambe et al. (2011). The concentrations of individual VOCs are estimated using the high time-resolution THC concentration and the distribution of VOCs in different phases of the driving cycle (see Tables S2-S4), and the OHR due to VOCs is obtained from these concentrations and respective reaction constants. The OH exposure in PAM was modelled at 20 second time interval for each driving cycle, and the average OH exposures for the cycles are presented in Table 1."**

The updated OH exposures also affect p.13 lines 12->:

**"In our experiments, the equivalent atmospheric age was approximately 3.9-6.2 hours during the CSUDC, when the most SOA formation took place. Thus our results are likely on the lower-end compared to maximum secondary aerosol formation potential, but similar OH exposures are reached as in SOA formation studies conducted with batch chambers (e.g. Platt et al. 2013; Gordon et al. 2014a; Nordin et al. 2013)."**

And in the conclusions, the references to low OH oxidation in case of E100 are removed:

**" For E100 no significant increase in secondary aerosol concentrations due to the cold start was observed ."**

**"For E10, the secondary aerosol formation was significantly larger than the primary PM emissions with secondary to primary PM –ratio of 13.4, whereas for E100 similar increase for PM mass after the PAM chamber was not observed. ."**

One interesting result from this work is the chemical composition of the exhaust as function of aging. While the authors touch upon this in Figure 6, it would seem that the AMS has a much better ability to determine well-known aerosol classes through

positive matrix factorization. Is this possible with this data set? Furthermore, is this possible as a function of time during a NEDC run, or is the time resolution not sufficient? Such work could greatly enhance the impact of this paper.

Yes, agreed that PMF would be a great addition. However, to our experience PMF analysis on this kind of data set is very difficult. Firstly, the due to the broad residence time distribution of the PAM, PMF analysis of data measured after the PAM is not possible, so PMF could be only used to primary emissions. In addition, the data is very spiky due to accelerations and decelerations, and not by changes in PM sources. Some averaging would likely be needed prior to applying PMF, which could compromise the time resolution. Therefore applying the PMF to this kind of data set is challenging. Also the focus of this article is to compare the different fuels and the changes in primary and secondary emissions, and PMF would not likely provide suitable data for that. Also, the article is quite long already. However, a PMF analysis of primary emissions for different fuels could be a topic for a separate article in the future.

SPECIFIC COMMENTS

Page 5, Line 20: Why was deionized water added to the E100 fuel to adjust water content to 4.4% m/m?

Water was considered to have a protecting role since neat alcohol can cause some potential risks towards an engine. The FFVs today are not warranted for use of 100% ethanol, and thus precaution is needed when using such in a research project.

Page 6, Line 5: Why was the PAM not placed after the secondary dilution system? It seems as if that would be the most realistic in terms of the partitioning of semi-volatile organic compounds.

Considering the modeling work done by Peng et al. (2015 and 2016), it would indeed be more realistic at least in terms of realistic photochemistry and avoiding the un-realistic photolysis of precursor gases, but these modeling results were not published at the time of the measurement. Also the partitioning would be more realistic with more dilution. On the other hand, the more dilution, the smaller the condensation sink and the larger the vapor wall-losses in the PAM chamber, as shown by Palm et al. (2016). Due to additional dilution, also the signal measured by the instruments would be smaller and thus more uncertain. If the secondary aerosol is formed by nucleation, the formed particles might be too small to detect for the AMS if the sample is diluted too much. In future measurements, the data published in this paper may help to pre-estimate the condensation sink and the concentrations of OH reactive gases in engine exhaust and thus help choosing the right dilution system.

Page 7, Line 13: What is the reason for this difference in collection efficiencies?

The previous studies have shown that the collection efficiency of an aerosol mass spectrometer is affected by particle losses (i) during transit through the inlet and lens, (ii) by particle beam divergence for both tungsten and laser vaporizers and by (iii) bounce effects from tungsten vaporizer (Matthew et al., 2008; Huffman et al., 2009; Onasch et al., 2012). It is known that in the standard AMS with only the tungsten vaporizer the CE can depend on the chemical composition and acidity of aerosol as well as sampling relative humidity (Middlebrook et al., 2012), the default value for the CE being 0.5. For the SP-AMS, the CE can vary significantly from the default value of 0.5 due to the laser vaporizer. Onasch et al. (2012) estimated collection efficiency of coated black particles in the SP-AMS to be 0.75, whereas Willis et al. (2014) measured CE=0.6 for bare regal

black (typically used as a surrogate for BC in laboratory) particles but they observed a significant increase in CE with increasing coating thickness.

Following sentence was added to the text to explain this: **Willis et al., 2014 demonstrated that also particle morphology affects the SP-AMS particle beam width that in turn affects the collection efficiency through the overlap of the particle beam and the laser beam.**

Page 7, line 15: What is the significance of 90 nm for agglomerates if the restriction set by the aerodynamic lens is 50 nm?

The sentence is poorly explained. Idea was to state that CE of gasoline soot is expected to be low due to its small particle size and morphology. Although the maximum of soot mode is at 90nm, a large portion of this mode is in smaller particles (below 50nm) which are poorly detected by SP-AMS. This chapter is reformulated to be:

The collection efficiency (CE) value, representing the fraction of sampled particle mass that is detected by the MCP detector, is required for the calculation of aerosol mass concentration measured by the AMS. The previous studies have shown that the collection efficiency of an aerosol mass spectrometer is affected by particle losses (i) during transit through the inlet and lens, (ii) by particle beam divergence for both tungsten and laser vaporizers and by (iii) bounce effects from tungsten vaporizer (Matthew et al., 2008; Huffman et al., 2009; Onasch et al., 2012). **Willis et al., (2014) demonstrated that also particle morphology affects the SP-AMS particle beam width that in turn affects the collection efficiency through the overlap of the particle beam and the laser beam.** Similar to Karjalainen et al. (2015) a CE = 1 was used in this study for all SP-AMS data. We acknowledge that it is likely that the collection efficiency might be underestimated for thinly coated, primary emissions whereas used CE = 1 is likely closer to correct value for than for heavily coated spherical secondary aerosol. **Also, we note that gasoline soot, consisting of agglomerates with average diameter below 90nm, will likely have low transmission efficiency in aerodynamic lens and thus might have lower collection efficiency than regal black.**

Page 7, line 15: Furthermore, does the soot vaporizer in the SP-AMS have a lower detection limit like the SP2? If so, that should be explicitly stated here.

This should not be an issue, SP-AMS should not have a lower detection limit than normal HR-ToF-AMS. According to Onasch et al., (2012) the $3\sigma$ detection limits for rBC mass concentration measurements are 0.26 µg·m−3 for 1s sampling and 0.03 µg·m−3 for 1 min collection. The organic and sulfate detection limits are 1.8 times higher and 0.2 times lower, respectively, in line with previously published results for HR-AMS instruments (DeCarlo et al. 2006). Following clarification was added to the text:

**For SP-AMS one second and 1 minute $3\sigma$ detection limits for submicrometer aerosol are <0.31 µg m-3 and < 0.03 µg m-3 for all species in the V-mode, respectively (DeCarlo et al., 2006, Onasch et al., 2012).**

Page 10, Line 24: Why are the CxHyOz fragments going down with PAM aging? Is this evidence of fragmentation?

This chapter was poorly formulated. It is revised based on reviewer comments. See answers for reviewers 1 and 2. Shortly, the figure shows contribution of these fragments. Although the contribution slightly decreases, the absolute mass increases for E10 and E85. Also must be noted that the contribution of oxidized hydrocarbons (CxHyOz + CxHyO) increases when compared to hydrocarbon contributions for all fuels.

Page 10, Line 30: Why are these losses not accounted for in this study?

See answer for reviewer #1. A chapter about particulate and vapor losses is added to the supplement.

Page 11, Line 28: The phrase "close to zero" has no physical meaning in this case. Please change to "below the detection limit of our fuel analysis" and indicate what that detection limit is.

Agreed. The sentence was reformulated to be: The main ions observed after PAM chamber were sulphate, nitrate and ammonium for all **fuels. In this study the sulphur contents of E10, E85 and E100 fuels were lower than 10 ppm according to the specifications EN 228, EN 15293 and EN 15376, respectively.**

Page 13, Line 33: If the total HC from GC, FT-IR, and HPLC, then why do the authors believe the yield is 30% lower than predicted?

All the theoretical SOA calculations were updated based on reviewer #1 comment. Now theoretical and measured values agree for E85.

Page 15, line 7-9: Why are small organic particles ruled out of this situation?

Based on reviewer #1 comments, the discussion about the size distributions was revised. Discussion about the nanoparticles was added to the text. See the answer for reviewer #1.

TECHNICAL COMMENTS

Page 2, Line 3: Do the authors mean to use the indefinite article "a" instead of the definite article "the" in front of New European Driving Cycle?

Yes, corrected.

Page 3, line 8: The sentence starting "In addition to : : :" is very long. The clarity of this sentence could be increased by splitting it into two sentences.

The sentence was split as suggested. "**In addition to primary PM, burning process in engine cylinder produces so called delayed primary aerosol. Delayed primary include species like sulfuric acid which are in tailpipe conditions in gaseous phase but will condense or nucleate immediately when the exhaust is cooled and diluted, without any significant chemical transformation in the atmosphere (Arnold et al., 2012; Rönkkö et al., 2013; Pirjola et al., 2015)."**

Page 3, Line 9: Do the authors mean the plural "processes" instead of the singular "process?"

Corrected to "processes".

Page 3, Line 11: There needs to be a "," after distribution.

Corrected as suggested.

Page 3, Line 11: What do authors mean here by "different processes?" Are they referring to primary vs delayed primary aerosol?

Yes. The sentence was clarified.

Page 3, Line 13: For clarity, the authors should describe how secondary emissions differ from delayed primary. This has not been described in the manuscript yet.

A following clarification was added to the end of the chapter. "**Difference between delayed primary and secondary emission is that secondary emissions form through different transformation processes in the atmosphere, whereas delayed primary emissions form in the cooling process without any significant chemical transformation due to external conditions such as UV, or atmospheric oxidants.**"

Page 3, Line 18: Are there references for the previous studies using flow through chambers? That would help put this work into a better context for the reader.

Discussion about the previous chamber studies is increased to the introduction chapter. However, we note that the amount of published emissions studies using a chamber is still limited.

Page 3, Line 20: There needs to be a "," after both instances of "chamber". Also, do the authors mean "Potential Aerosol Mass?" or "potential aerosol mass?"

Corrected as suggested.

Page 3, Line 23: Do the authors mean "secondary particulate emissions" or "Secondary Particulate Emissions."

Corrected to be "secondary particulate emissions".

Page 4, Line 1: Here, the authors have defined secondary processes, but they have mentioned it several times already. This statement would be the most effective after the first mention of secondary particulate matter.

Agreed. Sentence in question "Secondary particulate matter forms in the atmosphere via gas-to-particle conversion as oxidation process typically lowers the volatility (vapour pressure) of gaseous compounds." Moved to the end of first chapter of introduction.

Page 4, Line 5: The acronym FVV has not yet been defined.

Abbreviation was added to the manuscript.

Page 4, Line 3: For the sentence starting "E.g.," the use of "for example" in this sentence is redundant because E.g., is Latin for "for example."

E.g. from the beginning of sentence is removed.

Page 4, line 8: Since BTEX is being defined, the authors do not need to put "BTEX

=" in the parentheses. Furthermore, the right hand parentheses for this definition has
been omitted.

Ok. Corrected.

Page 4, Line 15: It is not clear how the last two sentences of this paragraph corroborate the statement that "different photo-oxidation pathways are also dependent on conditions." I would suggest either to delete these statements or add an additional statement that clarifies why this is important.

Sentences removed as suggested.

Page 4, Line 16: Do the authors mean to add the definite article "The" in front of
"European union?"

Yes, corrected as suggested.

Page 4, line 23: The authors should either place a "," before typically, or move it towards
the end of the sentence to read "hydrocarbon emissions are typically lower."

Corrected as suggested: "Previous studies have shown that primary PM, CO, HC, $NO_x$ and aromatic hydrocarbon emissions are typically lower for the E85 fuel.."

Page 5, Line 6: The acronym PAM has already been defined.

Corrected.

Page 5, Line 11: Do the authors mean "focused on" instead of "focused to?"

Corrected to be focused on.

Page 5, line 15: "First" after the semicolon does not need to be capitalized.

Corrected as suggested.

Page 5, line 15: I am confused what the urban driving cycle is as opposed to the NEDC, and how it is repeated twice if doing a cold start on a separate day is required.

A figure of NEDC speed profile is added to the supplement to clarify this. The idea is that the NEDC cycle (1180s) is split to three different parts. First 399 seconds represent typical urban driving (UDC), with cold start. The speed profile of second 400s is same as first one, but now the engine is warm i.e. cold start is required only once per cycle. Last part of NEDC cycle represents highway driving (EUDC). A chapter about the driving cycle is added to the supplement to clarify this and the description of the cycle is improved.

Page 5, line 15: This sentence is missing many definite articles. It should read "the urban," "the first," "the second," and "the last."

Corrected as suggested.

Page 6, Line 14: "-3" in "cm-3" should be a superscript.

Corrected.

Page 7, Line 1: What metals and elements specifically?

List of commonly detected metals and elements (Na, Al, Ca, V, Cr, Mn, Fe, Ni, Cu, Zn, Rb, Sr and Ba) added to the manuscript.

Page 7, Line 1: The phrase starting "and DeCarlo" is not an independent clause. Perhaps deleted the ";" and add a "," after "(2006)?"

Corrected as suggested.

Page 7, Line 31: The sentence starting "Total hydro carbons : : :" is entirely redundant because an almost exact replicate of this phrase was stated on Page 7, Line 27.

Sentence removed as unnecessary.

Page 7, Line 34: The sentence starting "In these measurements : : :" and the sentence that follows it on Page 8, Line 1 seems entirely out of place here. First, FTIR and HPLC measurements have not been mentioned yet. This sentence makes more sense if placed after the paragraph that runs from Page 8, Line 5 to Line 10. Finally, I believe the phrase "GC and HC" should be replaced by GC, FTIR, and HPLC" for this sentence to make sense.

Sentences were moved and corrected as suggested.

Page 8, Line 13: Delete "-" before "ratios."

Corrected.

Page 9, Line 24: Please enclose "2015" in parentheses.

Corrected.

Page 9, Line 26: Please enclose "2015" in parentheses.

Corrected.

Page 9, Line 26: "O:C ratio" is a redundant phrase as the : indicates that it is a ratio. Please remove " ratio" in this instance and all instances thereafter.

Corrected.

Page 12, Line 27: Do the authors mean "studied the exhaust of secondary?"

Sentence was reformulated to be: Suarez-Bertoa et al. (2015) has studied secondary aerosol formation potential of exhaust for vehicles using high ethanol content fuels (E75, E85).

Page 13, Line 13: There should be a "," after "Thus."
Corrected.

Page 14, Line 4: The phrase "for E10 largest" should read "for E10, the largest."
Corrected.

Page 14, Line 8: The definite article "the" should come before "largest."
Corrected.

Page 14, Line 9: The indefinite article "A" should come before "moderate." Furthermore,
the plus "were" should be changed to "was."
Corrected.

Page 14, Line 15: The sentence that starts "For E10 : : :" is missing commas and definite and indefinite articles. It should read
"For E10, the cold start had a dominating role in secondary aerosol formation, with a clear increase after the cold start : : :"
Corrected.

Page 14, Line 33: What does "ions" refer to in this case? Inorganic species, ions in the mass spectrum, or both?
Corrected to be inorganic ions.

Page 15, Line 6: Do the authors mean "dependent" instead of "depending?"
Yes, corrected as suggested.

---

## Author Response (AR2)

We thank the reviewers and co-editor for their time and comments. Here are the given minor comments in black text and our answers (in blue text) and changes made to manuscript (**in blue bold text**). We have responded to all the reviewer comments.

1. The figures and tables in the Supplement are not presented in order (e.g., Figure S6 is after Figure S22). This needs to be fixed.

All figure and table numbers in supplement are revised.

2. Section 3.5 - This section would be much more understandable if there was a direct comparison between measured and predicted SOA. I can find the Supplement tables and figures showing the SOA predictions, but these are not directly compared to the measurements. Thus it is not clear if the predicted SOA is larger or smaller than measured SOA for each of the fuels. Please revise the Supplement to make this clear.

Line 38 of page 12 states that the predicted SOA is shown in Figs S20-S22, but this does not seem to be the case. Those figures seem to show measurements after the PAM, but not predicted SOA. Please clarify.

Overall, I'm not sure how important this section is to the manuscript. If the goal of the section is to show that there is poor agreement between measured and predicted SOA when considering only VOC precursors, that seems outside of the scope of this manuscript and is a topic that has been covered extensively in other manuscripts. If the goal is to argue that observed SOA decreases for E85 and E100 because of lower aromatic emissions, the argument is not communicated clearly. Please clarify.

All figure and table numbers in supplement are revised. The predicted SOA is presented in Figs. S20-S22. The comparison between predicted and measured SOA is now shown in Fig. S23.

The aim of this section was to explain why the observed SOA formation decreases as the ethanol content increases. This objective is now clarified in Sect. 3.5 as follows:

"Using Eq. (1), **the measured VOCs and** previously measured yields for these VOCs,  **we can analyze why the SOA formation potential decreases as the ethanol content in the fuel increases.** We assumed that $\Delta HC$ equals the measured VOC concentration before PAM, and similarly to Platt et al., (2013), we use low-NOx yields to get an upper limit for SOA formation. The yields are listed in Table 4. For ethylbenzene, the SOA yield of m-xylene (0.38) was used (Ng et al. 2007; Platt et al. 2013). According to (Volkamer et al., 2009), acetylene ($C_2H_2$) SOA yield strongly depends on the liquid water content of aerosol. Here a value of 0.1 was assumed. The yields are corrected with corresponding wall -loss correction factors (Table 4) presented by Zhang et al. (2014).

The contribution of each measured VOC on predicted SOA is shown in supplementary (Tables S2-S4, Figs. S20-S22). **According to the predictions, the decrease in the SOA formation is caused by the decrease in aromatic compounds in the exhaust when the ethanol content in the fuel is increased. The comparison between the predictions and the measurements is shown in Fig. S23. The trend in predictions generally agree with the measurements except for E100, where the predicted SOA is higher than for E85. The predicted SOA for E100 mostly comes from acetylene (Fig. S10). Thus, the measured SOA formation potential seems to depend rather on the aromatic concentrations than on the acetylene.**

~~For E10, the predicted SOA mass was 5780 µg m-3 (4.08 mg km-1), for E85 800 µg m-3 (0.59 mg km-1) and for E100 1281 µg m-3 (0.94 mg km-1). The measured SOA for E10 was approximately the same as the predicted maximum SOA. In contrast, for E85 the measured SOA is 60 % lower than the predicted SOA, and for E100 no SOA formation was observed even though there are SOA precursors present in the exhaust gas.~~

~~The discrepancy between the predicted and measured SOA may result from the presence of NOx in the exhaust. The predicted SOA is an upper limit for SOA formation, based on the low-NOx yields, which are higher than the high-NOx yields (Table 4). Using the wall-loss corrected high-NOx yields, the predicted SOA emission factors are 1.37 mg km-1, 0.25 mg km-1 and 0.53 mg km-1 for E10, E85 and E100, respectively. Thus, the measured SOA for E10 is approximately 2.9 times higher than the predicted SOA using high-NOx yields, indicating that there are other VOCs or IVOCs (intermediate-volatility organic compounds) contributing to SOA formation than the measured ones, or that the NOx chemistry in PAM chamber is different than that of the smog chambers where the high-NOx yields are measured. For E85, the predicted SOA using high-NOx yields is approximately the same as the measured SOA.~~"

3. Page 4, last sentence in first paragraph (~line 15) - does the sentence about BTEX refer to BTEX emissions or oxidation of BTEX species in the atmosphere?

This sentence refers to chamber studies and theoretical studies about BTEX formation process during the aging process. We added clarification to this sentence: "E.g. for aromatic BTEX (BTEX = benzene, toluene, ethylbenzene, and xylenes; volatile organic compounds (VOC) typically found in petroleum derivates) compounds have been suggested to depend on, for example, prevailing NOx concentrations **during the aging process** (Andino et al., 1996;15 Hurley et al., 2001; Sato et al., 2007; Sato et al., 2012)."

4. Page 7, Line 28, sentence ending in "regal black" - please clarify that regal black is sometimes used to calibrate the SP-AMS. I don't think it was mentioned previously.

Yes, correct. This sentence was revised to contain this information. **"Also, we note that gasoline soot, consisting of agglomerates with average diameter below 90nm, will likely have low transmission efficiency in the aerodynamic lens and thus might have lower collection efficiency than regal black, that is typically used for calibration**

5. The discussion of reasons for higher observed rBC after the PAM in section 3.3.2 is improved. However, this section could benefit from being parsed into multiple paragraphs.

This section was split to two paragraphs. First paragraph describes the observation and the second one explores possible reasons for the observation.

6. The authors have measured the particle wall loss rate as a function of the particle size. The derived wall loss rates need to be applied to the measured raw particle distribution during each PAM experiment to yield the wall loss corrected organic aerosol masses.

Unfortunately, we don't have the information about the size of organic particles since the AMS did not measure the size-distributions, and thus the wall loss correction for the organic aerosol mass is difficult. In addition, the measured size-dependent particle wall loss rate does not directly apply to this case, because the particles are growing inside the PAM and thus the size is not constant as it is in the wall-loss measurement setup. However, we can estimate the wall loss by studying the HRLPI size-distribution. The total mass calculated from HRLPI number size-distribution increases by 9-16 % when the distribution is corrected for the wall losses (details in Table S5). This is now written at the end of Sect 2.2:

[revised manuscript text omitted]

Supplemental material includes:

**S1. Driving cycle and car preparation**

Cold-start tests were carried out by using the European exhaust emissions driving cycle, "NEDC" (Fig. S2), which is defined
5 in the UN ECE R83 regulation. NEDC totals 11.0 km and is divided into three test phases to study emissions at cold start and with warmed-up engines. The first (Cold start urban driving cycle, CSUDC) and second test phases (hot start urban driving cycle, HUDC) each consists of 2.026 km driving, and the third test phase, the extra-urban driving cycle (EUDC), is 6.955 km.

10 Preparation needs and stability issues related to the FFV cars were based in the earlier project (Aakko-Saksa et al., 2014). After the fuel change and prior to NEDC, a hot-start test was applied to monitor how warmed-up cars performed. For this purpose, the FTP75 city driving cycle was run as a hot-start test (FTP75 cold-start procedure is defined by the US Environmental Protection Agency EPA). FTP75 driving cycle totals 17.77 km, which is divided into three test phases including a 600 seconds pause. Before the FTP75 hot-start test, a "dummy" test FTP75 was conducted to stabilize cars for the actual hot-start test.
15 Thereby, preparation of cars before the cold-start NEDC test on the following test day to avoid carry-over effect was extensive. Two NEDC tests were conducted for each fuel. Table S1 includes the concentrations of regulated emissions (average ± st.dev) during the driving cycle.

**S2. OH reactivity**

The average OH reactivities (OHR) during different parts of the cycle for all fuels are presented in Tables S2-S4. The OHR for each compound is its concentration times the reaction rate constant with OH. The rate constants are taken from Atkinson and Arey (2003) and Li et al. (2015). The different parts of the driving cycle are defined as CSUDC (0-391 s), HUDC (392-787 s) and EUDC (788-1180 s).
25
**S3. PM and vapour losses in PAM chamber**

**S3.1 PM losses**

30 PAM chamber was designed with lower surface-area-to-volume (SA/V) ratio to minimize wall effects. Primary particle losses represented in Fig. S3, were measured in laboratory for a similar PAM chamber as was used in this study. Losses were in general quite small in the particle sizes that contains most of the aerosol mass: 25% at 50 nm, 15% at 100 nm and below 10% above 150 nm.

The particle number size distributions measured by HR-LPI were used to estimate how the particle losses in the PAM affect the measured total particle mass. If the measured HR-LPI number size distributions are corrected with the particle loss curve (Fig. S3), the total mass calculated from the number size distribution increases by 9-16 % depending on the phase of the cycle and the fuel (Table S5).

**S3.2**

**Vapour losses**

The secondary aerosol is formed when low volatility vapors condense on aerosols or form new particles. In the PAM chamber, these vapors may also condense onto walls, exit the chamber, or react with OH, which leads to fragmentation and increase in the saturation vapor pressure. Thus the potential aerosol mass is underestimated if these chamber related losses of low volatile vapors are not taken into account. We used the LVOC (low volatility organic compound) fate model presented by Palm et al. (2016) to estimate the losses of condensing organic vapors in the PAM chamber (model available at https://sites.google.com/site/pamwiki/hardware/estimation-equations). In the model, the relative fates of the vapor are estimated by studying the timescales of condensation on particles, condensation on chamber walls, reaction with OH radical and the residence time in the PAM chamber. Detailed description of the model can be found in Palm et al. (2016). Shortly: The lifetime for LVOC condensation is

$$\tau_{aer} = \frac{1}{4\pi \cdot CS \cdot D},$$

where D is the diffusion coefficient of the condensing molecule and CS is the condensational sink, which is calculated using the average of the HRLPI size distributions before and after PAM. The rate of LVOC loss to the walls is

$$\frac{1}{\tau_{wall}} = \frac{A}{V} \frac{2}{\pi} \sqrt{k_e D},$$

where A/V is the surface-area-to-volume ratio of the chamber, $k_e$ is the coefficient of eddy diffusion and D the diffusion coefficient.

The assumptions used in the model are same as in Palm et al. (2016): $D = 7 \times 10^{-6} \ m^2 s^{-1}$, $\alpha = 1$, mean free path $\lambda_g = 3\sqrt{\frac{\pi m_g}{8kT}} D \approx 1.173 \times 10^{-7} \ m$ (Pirjola et al. 1999), $\frac{A}{V} = 25 \ m^{-1}$ and $k_e = 0.0036 \ s^{-1}$.

The reaction rate constant for the reaction with OH is $k_{OH} = 1 \times 10^{-11} \ cm^3 \ molec.^{-1} \ s^{-1}$. LVOC is considered to fragment and form high-volatility molecules after five reactions with OH radical. Thus, the lifetime for fragmentation is

$$\tau_{OH} = \frac{5}{k_{OH} \cdot [OH]},$$

where [OH] is calculated based on the OH exposure and the residence time.

Using these lifetimes, the fate of LVOCs was calculated for each fuel and each part of the driving cycle, and the results are presented in Table S7. Because of the high condensational sink, over 95 % of the LVOCs condensed on aerosol in all cases according to this model. Thus, the chamber related losses of LVOCs are small.

Figures:

[Figure]

10    **Figure S1. Experimental setup (MFC = mass flow controller) used in this campaign. (Figure adapted from Karjalainen et al., 2016). with permission).**

[Figure]

**Figure S2: NEDC driving cycle**

[Figure]

**Figure S3. Primary particle loss ratio vses in a similar PAM chamber that was used in the study. (Figure adapted from Karjalainen et al., 2016 with permission).**

[Figure]

[Figure]

**Figure S4: The composition and concentration (mg/km) of emitted primary (a) and secondary (b) PM for each fuel.**

[Figure]

**Figure S5: Average mass spectra for primary emissions of E10 during the NEDC cycle.**

[Figure]

**Figure S6: Timeseries of primary organic, inorganic (sulfate, ammonium, nitrate, chloride) compounds and rBC during the NEDC cycle when using E10 fuel.**

[Figure]

**Figure S7. Average mass spectra over the NEDC cycle for E10 secondary emissions**

[Figure]

**Figure S8: Timeseries of PM organic, inorganic (sulfate, ammonium, nitrate, chloride) compounds and rBC observed after PAM chamber during the NEDC cycle when using E10 fuel.**

[Figure]

**Figure S915: Average mass spectra over the NEDC cycle for E85 primary emissions**

[Figure]

**Figure S106: Timeseries of primary organic, inorganic (sulfate, ammonium, nitrate, chloride) compounds and rBC during the NEDC cycle when using E85 fuel.**

[Figure]

**Figure S117: Average mass spectra over the NEDC cycle for E85 secondary emissions**

[Figure]

**Figure S128: Timeseries of organic, inorganic (sulfate, ammonium, nitrate, chloride) compounds and rBC measured after PAM chamber during the NEDC cycle when using E85 fuel.**

[Figure]

**Figure S19: Average mass spectra over the NEDC cycle for E100 secondary emissions**

[Figure]

**Figure S1420: Timeseries of organic, inorganic (sulfate, ammonium, nitrate, chloride) compounds and rBC after PAM chamber during the NEDC cycle when using E100 fuel.**

[Figure]

**Figure S15: Average mass spectra over the NEDC cycle for E100 primary emissions**

[Figure]

5   **Figure S16: Timeseries of organic, inorganic (sulfate, ammonium, nitrate, chloride) compounds and rBC during the NEDC cycle when using E100 fuel.**

[Figure]

**Fig S17. With PAM chamber to w/o PAM chamber ratios for C$_2$-C$_5$ fragments for E10.**

[Figure]

**Figure S18. Average number size distributions measured for different fuels (E10, E85, E100) with and without PAM chamber.**

[Figure]

**Figure S19. Average volume size distributions measured for different fuels (E10, E85, E100) with and without PAM chamber. Note the scaling by a factor 0.1 for E10 size distribution.**

[Figure]

**Figure S20. Predicted SOA during CSUDC for E10 fuel and low-NOx yields.**

[Figure]

**Figure S21. Predicted SOA during CSUDC for E85 fuel and low-NOx yields..**

[Figure]

**Figure S22. Predicted SOA during CSUDC for E100 fuel and low-NOx yields..**

[Figure]

**Figure S23. Comparison between measured and predicted SOA formation potential. The predictions are based on VOC measurements and SOA yields as described in Sect. 3.5.**

**Tables:**

**Table S1. Emissions (average ± st.dev) over the cold-start European test cycle (mg/km).**

|  | Fuel | CO | HC | NOx | PM | CO2 |
|---|---|---|---|---|---|---|
| **Concentration** | E10 | 396.6 | 30.4 | 43.3 | 1.4 | 174 181 |
|  | E85 | 142.1 | 29.9 | 30.3 | 1.3 | 165 837 |
|  | E100 | 368.2 | 192.9 | 31.7 | 0.9 | 165 196 |
| **Standard deviation** |  |  |  |  |  |  |
|  | E10 | ±80.52 | ±10.02 | ±2.76 | ±0.08 | 2 729 |
|  | E85 | ±15.33 | ±2.96 | ±1.00 | ±0.64 | 727 |
| **---** | E100 | ±187.57 | ±109.4 | ±16.51 | ±0.01 | 1 292 |

**Table S2. OHR and predicted SOA for E10 fuel.**

[revised manuscript text omitted]

--

| | Rate constant | Concentration | | | OHR | | | Predicted SOA (Low-NOx) | | | | Predicted SOA (High-NOx) | | | |
|---|---|---|---|---|---|---|---|---|---|---|---|---|---|---|---|
| | | CSUDC | HUDC | EUDC | CSUDC | HUDC | EUDC | Yield | CSUDC | HUDC | EUDC | Yield | CSUDC | HUDC | EUDC |
| | cm3 molec.-1 s-1 | molec. cm-3 | molec. cm-3 | molec. cm-3 | s-1 | s-1 | s-1 | | mg km-1 | mg km-1 | mg km-1 | | mg km-1 | mg km-1 | mg km-1 |
| Methane | 6.40E-15 | 2.93E+14 | 2.61E+13 | 9.16E+12 | 2 | 0 | 0 | 0.00 | 0.00 | 0.00 | 0.00 | 0.00 | 0.00 | 0.00 | 0.00 |
| Ethane | 2.48E-13 | 0.00E+00 | 0.00E+00 | 0.00E+00 | 0 | 0 | 0 | 0.00 | 0.00 | 0.00 | 0.00 | 0.00 | 0.00 | 0.00 | 0.00 |
| Ethene | 8.52E-12 | 1.03E+14 | 0.00E+00 | 0.00E+00 | 880 | 0 | 0 | 0.00 | 0.00 | 0.00 | 0.00 | 0.00 | 0.00 | 0.00 | 0.00 |
| Propane | 1.09E-12 | 0.00E+00 | 0.00E+00 | 0.00E+00 | 0 | 0 | 0 | 0.00 | 0.00 | 0.00 | 0.00 | 0.00 | 0.00 | 0.00 | 0.00 |
| Propene | 2.63E-11 | 1.46E+12 | 0.00E+00 | 0.00E+00 | 38 | 0 | 0 | 0.00 | 0.00 | 0.00 | 0.00 | 0.00 | 0.00 | 0.00 | 0.00 |
| Acetylene | 8.80E-13 | 2.07E+13 | 8.74E+11 | 3.55E+11 | 18 | 1 | 0 | 0.10 | 1.11 | 0.04 | 0.01 | 0.10 | 1.11 | 0.04 | 0.01 |
| Iso-butene | 5.14E-11 | 0.00E+00 | 0.00E+00 | 0.00E+00 | 0 | 0 | 0 | 0.00 | 0.00 | 0.00 | 0.00 | 0.00 | 0.00 | 0.00 | 0.00 |
| 1,3-Butadiene | 6.66E-11 | 0.00E+00 | 0.00E+00 | 0.00E+00 | 0 | 0 | 0 | 0.18 | 0.00 | 0.00 | 0.00 | 0.18 | 0.00 | 0.00 | 0.00 |
| Benzene | 1.22E-12 | 6.05E+11 | 3.12E+11 | 1.01E+11 | 1 | 0 | 0 | 0.67 | 0.64 | 0.27 | 0.06 | 0.35 | 0.34 | 0.14 | 0.03 |
| Toluene | 5.63E-12 | 7.26E+11 | 5.20E+11 | 9.62E+10 | 4 | 3 | 1 | 0.57 | 0.78 | 0.45 | 0.06 | 0.15 | 0.20 | 0.12 | 0.02 |
| Ethylbenzene | 7.00E-12 | 0.00E+00 | 1.12E+12 | 0.00E+00 | 0 | 8 | 0 | 0.67 | 0.00 | 1.31 | 0.00 | 0.35 | 0.00 | 0.69 | 0.00 |
| m/p-xylene | 1.87E-11 | 0.00E+00 | 0.00E+00 | 0.00E+00 | 0 | 0 | 0 | 0.68 | 0.00 | 0.00 | 0.00 | 0.10 | 0.00 | 0.00 | 0.00 |
| o-xylene | 1.36E-11 | 0.00E+00 | 0.00E+00 | 0.00E+00 | 0 | 0 | 0 | 0.10 | 0.00 | 0.00 | 0.00 | 0.05 | 0.00 | 0.00 | 0.00 |
| CO | 2.37E-13 | 1.30E+15 | 4.85E+13 | 1.77E+14 | 309 | 12 | 42 | 0.00 | 0.00 | 0.00 | 0.00 | 0.00 | 0.00 | 0.00 | 0.00 |
| Formaldehyde | 9.37E-12 | 1.19E+13 | 1.15E+12 | 7.52E+11 | 111 | 11 | 7 | 0.00 | 0.00 | 0.00 | 0.00 | 0.00 | 0.00 | 0.00 | 0.00 |
| Acetaldehyde | 1.50E-11 | 8.24E+13 | 5.05E+11 | 4.23E+11 | 1237 | 8 | 6 | 0.00 | 0.00 | 0.00 | 0.00 | 0.00 | 0.00 | 0.00 | 0.00 |
| Ethanol | 2.90E-12 | 2.90E+14 | 8.42E+12 | 6.46E+12 | 841 | 24 | 19 | 0.00 | 0.00 | 0.00 | 0.00 | 0.00 | 0.00 | 0.00 | 0.00 |
| | | | | SUM | 3441 | 66 | 75 | | 2.53 | 2.06 | 0.14 | | 1.65 | 0.98 | 0.06 |
| | | | | Total predicted SOA | | | | | 0.94 mg km-1 | | | | 0.53 mg km-1 | | |

**Table S5: Detection limits as a ppm and mg/km for compounds measured with the FTIR.**

| | Detection limit | |
|---|---|---|
| | Concentration at 1-s interval (ppm) | European test (mg/km) |
| **Carbon monoxide (CO)** | 7 | 8 |
| **Nitric oxide (NO)** | 13 | 15 |
| **Nitrogen dioxide (NO$_2$)** | 2/10 | 4 |
| **Nitrous oxide (N$_2$O)** | 4 | 4 |
| **Ammonia** | 2 | 1 |
| **Methanol** | 2 | 1 |
| **Ethanol** | 4 | 7 |
| **Isobutanol** | 3 | 9 |
| **n-Butanol** | 4 | 12 |
| **ETBE** | 2 | 8 |
| **Formaldehyde** | 5 | 6 |

| Acetaldehyde | 5 | 9 |

**Table S6: Increase in the HRLPI mass due to PAM particle wall-loss correction**

| Fuel | CSUDC | HUDC | EUDC |
|------|-------|------|------|
| E10 | 9 % | 13 % | 13 % |
| E85 | 13 % | 16 % | 16 % |
| E100 | 15 % | 16 % | 16 % |

10 **Table S7: LVOC fate in the PAM chamber.**

| | E10 | | | E85 | | | E100 | | |
|---|---|---|---|---|---|---|---|---|---|
| | CSUDC | HUDC | EUDC | CSUDC | HUDC | EUDC | CSUDC | HUDC | EUDC |
| Condense on aerosol | 99.3 % | 98.6 % | 98.6 % | 97.1 % | 95.5 % | 95.6 % | 95.2 % | 95.6 % | 95.7 % |
| Condense on walls | 0.5 % | 0.5 % | 0.5 % | 2.3 % | 2.7 % | 2.6 % | 3.7 % | 2.4 % | 2.0 % |
| Fragmentation | 0.2 % | 0.9 % | 0.9 % | 0.6 % | 1.8 % | 1.8 % | 0.7 % | 2.0 % | 2.3 % |
| Exit the chamber | 0.0 % | 0.0 % | 0.0 % | 0.0 % | 0.0 % | 0.0 % | 0.3 % | 0.0 % | 0.0 % |

---

## Author Response (AR3)

Author's response:

**Co-Editor Decision: Publish subject to technical corrections** (24 Mar 2017)
Comments to the Author: The authors have addressed the referee comments adequately, and I recommend publication subject to one minor technical correction: In the correction to page 7, line 28, please change 'that' to 'which' in the phrase "and thus might have lower collection efficiency than regal black, that is typically used for calibration"

Sentence corrected as suggested by editor:
**"Also, we note that gasoline soot, consisting of agglomerates with average diameter below 90nm, will likely have low transmission efficiency in the aerodynamic lens and thus might have lower collection efficiency than regal black, which is typically used for calibration."**